# Neuropeptide Bursicon and its receptor-mediated the transition from summer-form to winter-form of *Cacopsylla chinensis*

**Zhixian Zhang[1], Jianying Li[1], Yilin Wang[1], Zhen Li[1], Xiaoxia Liu[1], Songdou Zhang[1,2]***

[1]Department of Entomology and MOA Key Lab of Pest Monitoring and Green Management, College of Plant Protection, China Agricultural University, Beijing, China; [2]Sanya Institute of China Agricultural University, Sanya City, China

## eLife Assessment

This **important** study reports that the neurohormone, bursicon, and its receptor, play a role in the seasonal polyphenism of the bug Cacopsylla chinensis. Low temperature activates the bursicon signaling pathway during the transition from the summer to the winter form, affecting cuticle pigment and thickness as well as chitin content. The **solid** experiments reveal how bursicon signaling, which is modulated by the microRNA miR-6012, regulates features of polyphenism related to the exoskeleton, although it is less clear what the upstream regulatory events are.

*For correspondence:
sdzhang2013@cau.edu.cn

**Competing interest:** The authors declare that no competing interests exist.

**Abstract** Seasonal polyphenism enables organisms to adapt to environmental challenges by increasing phenotypic diversity. *Cacopsylla chinensis* exhibits remarkable seasonal polyphenism, specifically in the form of summer-form and winter-form, which have distinct morphological phenotypes. Previous research has shown that low temperature and the temperature receptor *CcTRPM* regulate the transition from summer-form to winter-form in *C. chinensis* by impacting cuticle content and thickness. However, the underling neuroendocrine regulatory mechanism remains largely unknown. Bursicon, also known as the tanning hormone, is responsible for the hardening and darkening of the insect cuticle. In this study, we report for the first time on the novel function of Bursicon and its receptor in the transition from summer-form to winter-form in *C. chinensis*. Firstly, we identified *CcBurs-α* and *CcBurs-β* as two typical subunits of Bursicon in *C. chinensis*, which were regulated by low temperature (10 °C) and *CcTRPM*. Subsequently, *CcBurs-α* and *CcBurs-β* formed a heterodimer that mediated the transition from summer-form to winter-form by influencing the cuticle chitin contents and cuticle thickness. Furthermore, we demonstrated that *CcBurs-R* acts as the Bursicon receptor and plays a critical role in the up-stream signaling of the chitin biosynthesis pathway, regulating the transition from summer-form to winter-form. Finally, we discovered that miR-6012 directly targets *CcBurs-R*, contributing to the regulation of Bursicon signaling in the seasonal polyphenism of *C. chinensis*. In summary, these findings reveal the novel function of the neuroendocrine regulatory mechanism underlying seasonal polyphenism and provide critical insights into the insect Bursicon and its receptor.

## Introduction

Polyphenism is a transformation phenomenon of phenotypic plasticity, where a single genome produces multiple distinct phenotypes in response to environmental cues (*Simpson et al., 2011*).

**eLife digest** A bug known as pear psylla is a common pest on pear trees in China and other East Asian countries. It feeds off the sap of young leaves and shoots, causing damage to the trees and decreasing the amount of fruit they produce. To survive all-year round, pear psylla changes its body between distinct summer- and winter-forms, a phenomenon known as seasonal polyphenism. In winter, cooler temperatures trigger young pear psylla to darken and thicken the protective cuticle layer coating their bodies. However, the mechanisms behind this seasonal transformation are not fully understood.

One possible regulator of this process is the hormone Bursicon which is known to control cuticle development in juvenile insects. The hormone has two subunits that join to form dimers, which then activate specific receptors that initiate signaling pathways within the insect's body. Here, Zhang et al. used molecular biology and genetic techniques to study the role of Bursicon dimers in seasonal polyphenism in pear psylla.

Bursicon can assemble either as a homodimer (made up of two identical subunits), or a heterodimer (made up two different subunits). Zhang et al. found that low temperatures triggered the formation of both homodimers and heterodimers of Bursicon. However, only the heterodimers activated a receptor, called *CcBurs-R*, which enabled the pear psylla to transition into their winter-form. The team also identified a small molecule called a micro RNA that regulates this switch by decreasing the production of the *CcBurs-R* receptor.

The findings by Zhang et al. advance our understanding of how seasonal polyphenism operates in pear psylla. Many other insects display seasonal polyphenism, and further research could reveal whether Bursicon plays a similar regulatory role across different species.

In recent years, polyphenism has garnered increasing attention and has become a focal point of research, such as ecology, evolutionary biology, epigenetics, and entomology. Nature presents us with numerous remarkable examples of polyphenism. For instance, we observe seasonal polyphenism in psylla (*Butt and Stuart, 1986*) and butterflies (*Daniels et al., 2014*; *Baudach and Vilcinskas, 2021*), sexual and wing polyphenism in aphids and planthoppers (*Xu et al., 2015*; *Shang et al., 2020*), caste polyphenism in ants and honeybees (*Kucharski et al., 2008*; *Bonasio et al., 2012*), sex determination in reptiles and fish regulated by temperature and social factors (*Janzen and Phillips, 2006*; *Liu et al., 2017*), and environmentally induced polyphenism in plants (*Gratani, 2014*). Undoubtedly, polyphenism plays a major contributor to the population dynamics of insects worldwide (*Noor et al., 2008*). Numerous studies have reported that insect polyphenism is influenced by a range of external environment factors, such as temperature, population density, photoperiod, and dietary nutrition (*Simpson et al., 2011*; *Ma et al., 2011*; *An et al., 2012a*; *Zhang et al., 2019*). Additionally, internal neuro-hormones, including insulin, dopamine, and ecdysone, have been found to play crucial roles in insect polyphenism (*Ma et al., 2011*; *Uehara et al., 2011*; *Xu et al., 2015*; *Vellichirammal et al., 2017*). However, the specific molecular mechanism underling temperature-dependent polyphenism still requires further clarification.

*Cacopsylla chinensis* (Yang & Li) is a pear psylla belonging to the Hemiptera order, which causes severe damage to trees and fruits in the major pear production areas across East Asian countries, including China and Japan (*Hildebrand et al., 2000*; *Wei et al., 2020*). This phloem-sucking psylla inflicts harm on young shoots and leaves in both adult and nymph stages, leading to stunted and withered pear trees (*Ge et al., 2019*). Furthermore, *C. chinensis* secretes a substantial amount of honeydew and acts as a vector for plant pathogenic microorganisms, such as the phytoplasma of pear decline disease and *Erwinia amovora* (*Hildebrand et al., 2000*). Importantly, this pest demonstrates strong adaptability to its environment and exhibits seasonal polyphenism, manifesting as summer-form (SF) and winter-form (WF), which display significant differences in morphological characteristics throughout the seasons (*Ge et al., 2019*; *Zhang et al., 2023*). The summer-form has a lighter body color and causes more severe damage, while the winter-form, in contrast, has a brown to dark brown body color, a larger body size, and stronger resistance to weather conditions (*Ge et al., 2019*; *Tougeron et al., 2021*). In a previous study, Zhang et al. demonstrated that a low temperature of 10 °C and the temperature receptor *CcTRPM* regulate the transition from summer-form to winter-form in *C.*

*chinensis* by affecting cuticle thickness and chitin content (*Zhang et al., 2023*). Up to now, no insect hormones or neuropeptides underling this seasonal polyphenism in *C. chinensis* have been identified.

Bursicon, also known as the tanning hormone, was initially discovered in the 1960s through neck-ligated assays. It serves a highly conserved function in insects by inducing the clerotization and melanization of the new cuticle in larvae and facilitating wing expansion in adults (*Dewey et al., 2004*). Bursicon is a heterodimer neuropeptide composed of two subunits, Bursicon-α and Bursicon-β, which exert their effects through the leucine-rich repeats-containing G- protein-coupled receptor, also known as the Bursicon receptor (*Luo et al., 2005*). In *Drosophila*, flies with mutated Bursicon receptor, such as the *rk* gene, or deficient in one of the Bursicon subunits, exhibit improper tanning and altered body shape (*Luan et al., 2006*). Similarly, in the model insect *Tribolium castaneum*, RNA interference experiments have demonstrated that the Bursicon receptor (*Tcrk*) is not only required for cuticle tanning, but also crucial for the development and expansion of integumentary structures (*Bai and Palli, 2010*). Interestingly, it has been reported that Bursicon homodimers can activate the NF-kB transcription factor *Relish*, leading to the induction of innate immune and stress genes during molting (*An et al., 2012b*). Consequently, insects exposed to cold conditions exhibit larger body size and darker cuticular melanization than those reared in warmer environments (*Shearer et al., 2016*). Given this background, Bursicon and its receptor are expected to play a significant role in the seasonal polyphenism of *C. chinensis*.

MicroRNAs (miRNAs), which are approximately 23 nucleotides in length and belong to a class of small noncoding RNAs, play a crucial role in the regulation of posttranscriptional gene expression (*Lucas and Raikhel, 2013*). Increasing studies have shown that miRNAs are important in insect poly-phenism, such as miR-31, miR-9, and miR-252, as well as hormone signaling, for example, miR-133 in dopamine synthesis (*Yang et al., 2014*; *Zhang et al., 2020*; *Shang et al., 2020*; *Zhang et al., 2023*). However, there have been no reports on miRNAs targeting Bursicon and its receptor. There-fore, studying the molecular mechanism of miRNA regulation of the Bursicon receptor at the post-transcriptional level would be highly innovation. In this study, we conducted bioinformatics analysis, qRT-PCR, and western blot to identify two Bursicon subunits (*CcBurs-α* and *CcBurs-β*) and their associ-ation with low temperature of 10 °C. We then employed RNAi, cuticle staining, and transmission elec-tron microscopy to study the effects of *CcBurs-α* and *CcBurs-β* on cuticle content, cuticle thickness, and the transition percent from summer-form to winter-form in *C. chinensis*. Furthermore, we identi-fied *CcBurs-R* as the Bursicon receptor and investigated its role in the transition from summer-form to winter-form. Finally, through in vivo and in vitro assays, we discovered that miR-6012 targets *CcBurs-R* and is involved in the seasonal polyphenism. These efforts not only shed light on the novel function of Bursicon and its receptor in mediating the transition from summer-form to winter-form in *C. chinensis*, but also enhance our understanding of the neuroendocrine basis of insect seasonal polyphenism.

## Results

### Investigation of the relationship between nymph phenotype, cuticle pigment absorbance, and cuticle thickness during the transition from summer-form to winter-form in *C. chinensis*

In *C. chinensis*, exposure to a low temperature of 10 °C triggers the activation of the temperature receptor *CcTRPM*, leading to the transition from summer-form to winter-form by influencing cuticle tanning and cuticle thickness (*Zhang et al., 2023*). To elucidate the association between these param-eters and the cuticle tanning threshold, we investigated nymph phenotypes, cuticle pigment absor-bance levels, and cuticle thickness in *C. chinensis* over varying time intervals (3, 6, 9, 12, 15 days) under either 10 °C or 25 °C temperature conditions. Nymphs exhibited a light yellow and transparent hue at 3, 6, and 9 days, while those at 12 and 15 days displayed shades of yellow-green or blue-yellow under 25 °C conditions. Conversely, under 10 °C conditions, nymphs darkened at the abdomen's end at 3, 6, and 9 days, developed numerous light black stripes on their thorax and abdomen at 12 days, and presented an overall black-brown appearance with dark brown stripes on the left and right sides of each thorax and abdominal section at 15 days (*Figure 1—figure supplement 1A*). Notably, the dorsal and ventral sides of the abdomen exhibited a pronounced black-brown coloration at 10 °C. The UV absorbance of the total pigment extraction at a wavelength of 300 nm significantly increased following 10 °C exposure for 3, 6, 9, 12, and 15 days compared to the 25 °C treatment group (*Figure 1—figure*

*supplement 1B*). Moreover, cuticle thicknesses exhibited an increase following 10 °C exposure for 3, 6, 9, 12, and 15 days compared to the 25 °C treatment group (*Figure 1—figure supplement 1C*).

## Molecular identification of *CcBurs-α* and *CcBurs-β* in *C. chinensis*

Sequence analysis showed that the open reading frame (ORF) of *CcBurs-α* (GenBank: OR488624) is 480 bp long, encoding a predicted polypeptide of 159 amino acids. The polypeptide had a molecular weight of 17.45 kDa and a theoretical isoelectric point (*pI*) of 6.13. The complete ORF of *CcBurs-β* (GenBank: OR488625) is 405 bp, encoding a polypeptide of 134 amino acid residues. The predicated molecular weight of *CcBurs-β* was 15.21 kDa and a theoretical *pI* of 5.24. Amino acid sequence alignment analysis revealed that CcBurs-α and CcBurs-β shared high amino acid identity with homologs from other selected insect species (*Figure 1A and B*). Both subunits contained eleven conserved cysteine residues, marked with red stars. Phylogenetic analysis (*Figure 1—figure supplement 2A–B*) indicated that *CcBurs-α* was most closely related to the *DcBurs-α* homologue (*Diaphorina citri*, XP_008468249.2), while *CcBurs-β* was most closely related to *DcBurs-β* (*D. citri*, AWT50591.1) among the selected species. The potential tertiary protein structure and molecular docking of *CcBurs-α* and *CcBurs-β* were constructed using the Phyre$^2$ server and PyMOL-v1.3r1 software (*Figure 1C*). To investigate the identities of homodimers and heterodimers of *CcBurs-α* and *CcBurs-β*, SDS-PAGE with reduced and non-reduced gels was used. When expressed as individual subunits, they formed α+α and β+β homodimers, as the molecular size of α or β doubled in the non-reduced gel compared to the reduced gel (*Figure 1D*). When co-expressed, most α and β subunits formed the *CcBurs-α+β* heterodimer (*Figure 1D*).

The temporal expression profile revealed that both *CcBurs-α* and *CcBurs-β* were ubiquitous in all developmental stages, with lower expression in eggs and nymphs and higher expression in adults of both summer-form and winter-form (*Figure 1—figure supplement 3C–D*). Increased gene expression levels may potentially contribute to the transition from summer-form to winter-form in *C. chinensis*. Spatially,,*CcBurs-α* and *CcBurs-β* were detected in all investigated nymph tissues and were expressed most prominently in the head (*Figure 1—figure supplement 3E–F*). In addition to the midgut, both *CcBurs-α* and *CcBurs-β* also showed high expression in other selected tissues of the winter-form compared to the summer-form, especially in the head and cuticle, except the midgut. Results from temperature treatment exhibited that the mRNA expression of *CcBurs-α* and *CcBurs-β* significantly increased after 10 °C treatment for 3, 6, and 10 days compared to 25 °C treatment (*Figure 1E and F*). Meanwhile,,qRT-PCR results indicated that the transcription levels of both *CcBurs-α* and *CcBurs-β* were noticeably down-regulated after successful knockdown of the temperature receptor *CcTRPM* by RNAi at 3, 6, and 10 days (*Figure 1G–H*, *Figure 1—figure supplement 4*). These data suggest that *CcBurs-α* and *CcBurs-β* are regulated by a low temperature of 10 °C and *CcTRPM*, and may serve as down-stream signals involved in the seasonal polyphenism of *C. chinensis*.

## *CcBurs-α* and *CcBurs-β* were essential for the transition from summer-form to winter-form

To investigate the role of *CcBurs-α* and *CcBurs-β* in the transition from summer-form to winter-form of *C. chinensis*, newly hatched first instar nymphs of summer-form were fed with dsCcBurs-α, dsCcBurs-β, or dsEGFP. qRT-PCR results exhibited that feeding dsCcBurs-α or dsCcBurs-β extremely reduced the expression of the target gene under 10 °C conditions. The RNAi efficiencies of *CcBurs-α* and *CcBurs-β* were approximately 66–78% and 69–79% at 3, 6, and 10 days compared to dsEGFP feeding (*Figure 2A and B*).

After successful knockdown of *CcBurs-α*,*CcBurs-β*, or both, the UV absorbance of total pigment extraction at a wavelength of 300 nm in dsCcBurs-α-treated (0.18), dsCcBurs-β-treated (0.19), and dsCcBurs-α+β-treated (0.07) nymphs was dramatically lower than that in dsEGFP-treated nymphs (0.85) under 10 °C condition (*Figure 2C*). This finding indicates that *CcBurs-α* and *CcBurs-β* play a prominent role in cuticle pigment formation in the winter-form in *C. chinensis*. Moreover, both the results of cuticle chitin content determination and cuticle ultrastructure observation indicated that knockdown of *CcBurs-α*,*CcBurs-β*,or both markedly reduced the cuticle chitin content (about 0.33, 0.32, 0.14) and cuticle thicknesses (about 1.44, 1.53, 0.73 μm) compared with dsEGFP-treated nymphs (1.00, 3.39 μm) under 10 °C condition, respectively (*Figure 2D–G*). Interestingly, the results of pigmentation absorbance and cuticle thickness after *CcBurs-α* or *CcBurs-β* knockdown were similar to

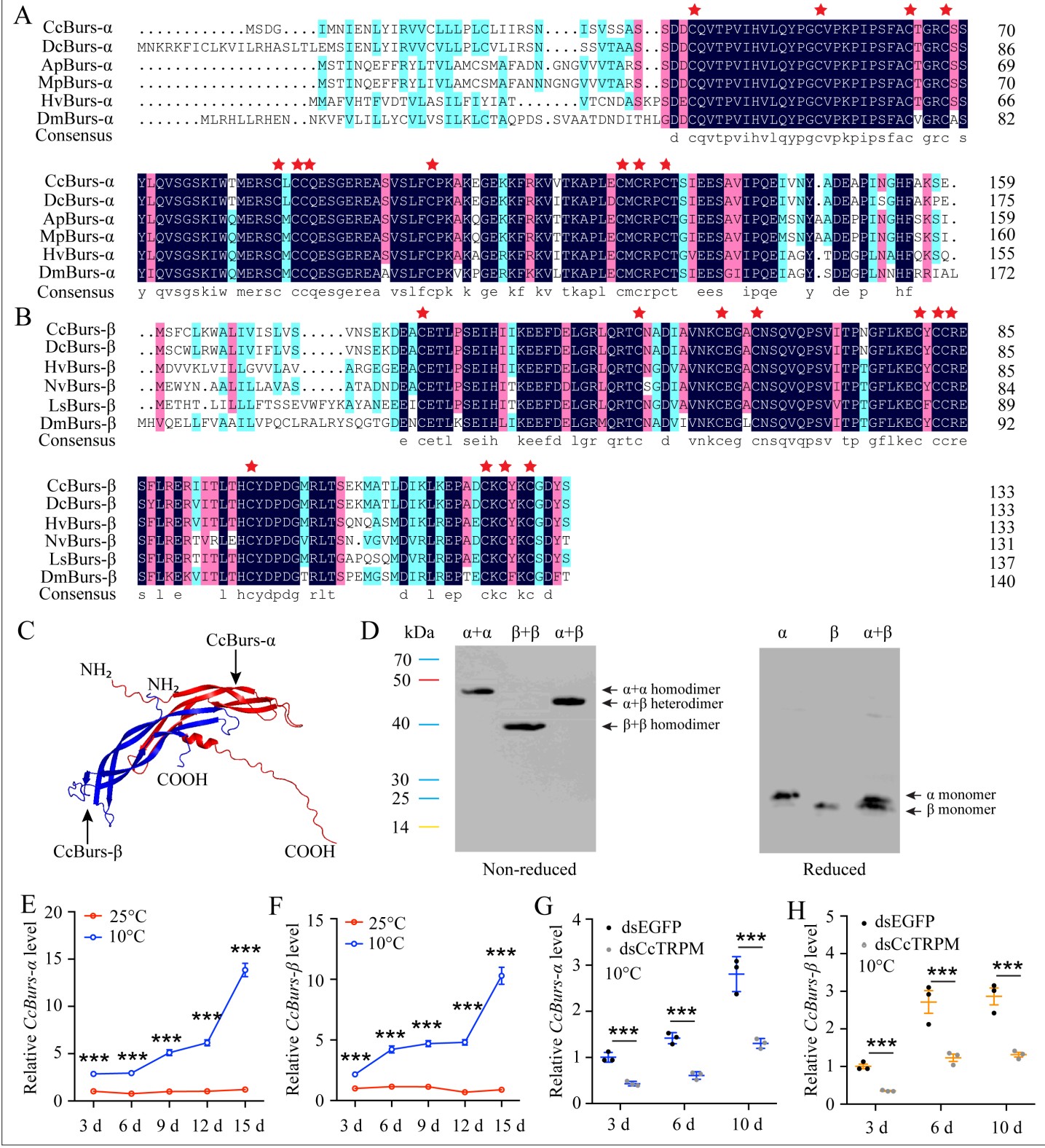

**Figure 1.** Molecular characteristic of *CcBurs*-α and *CcBurs*-β in *C. chinensis*. (**A**) Multiple alignments of the amino acid sequences of *CcBurs*-α with homologs from five other insect species. Black represents 100% identity, red represents 75% identity, and blue represents <75% identity. *CcBurs*-α (*C. chinensis*, OR488624), *DcBurs*-α (*Diaphorina citri*, XP_008468249.2), *ApBurs*-α (*Acyrthosiphon pisum*, XP_001946341.1), *MpBurs*-α (*Myzus persicae*, XP_022171710.1), *HvBurs*-α (*Homalodisca vitripennis*, XP_046670477.1), *DmBurs*-α (*Drosophila melanogaster*, CAH74223.1). The corresponding GenBank accession numbers are as follows. (**B**) Multiple alignments of the amino acid sequences of *CcBurs*-β with homologs from five other insect

*Figure 1 continued on next page*

*Figure 1 continued*

species. Black represents 100% identity, red represents 75% identity, and blue represents <75% identity. *CcBurs-β* (*C. chinensis*, OR488625), *DcBurs-β* (*D. citri*, AWT50591.1), *HvBurs-β* (*H. vitripennis*, XP_046671521.1), *NvBurs-β* (*Nezara viridula*, AZC86173.1), *LsBurs-β* (*Laodelphax striatellus*, AXF48186.1), *DmBurs-β* (*D. melanogaster*, CAH74224.1). The corresponding GenBank accession numbers are as follows. (**C**) Predicted protein tertiary structures of *CcBurs-α* and *CcBurs-β*. (**D**) Western blot analysis of Bursicon proteins using anti-His-Tag antibody with non-reduced and reduced SDS-PAGE. The left numbers indicate the positions of pre-stained protein markers. Lanes of α, β, and α+β represent separate expression of CcBurs-α, CcBurs-β, or co-expressed of α+β. Monomers were not present under non-reduced conditions. (**E-F**) Relative mRNA expression of *CcBurs-α* and *CcBurs-β* after 25 °C or 10 °C treatments at 3, 6, 9, 12, and 15 days (n=3). (**G-H**) Effect of temperature receptor *CcTRPM* knockdown on the mRNA expression of *CcBurs-α* and *CcBurs-β* at 3, 6, and 10 days under 10 °C condition (n=3). Data in 1E-1H are shown as the mean ± SE with three independent biological replications, with at least 50 nymphs for each biological replication. Statistically significant differences were determined using pair-wise Student's *t*-test in SPSS 26.0 software, and significance levels were denoted by ***p<0.001.

The online version of this article includes the following source data and figure supplement(s) for figure 1:

**Source data 1.** Labelled file for the western blot analysis in *Figure 1D* (reduced gel).

**Source data 2.** Original, uncropped file for the western blot analysis in *Figure 1D* (reduced gel).

**Source data 3.** Labelled file for the western blot analysis in *Figure 1D* (non-reduced gel).

**Source data 4.** Original, uncropped file for the western blot analysis in *Figure 1D* (non-reduced gel).

**Figure supplement 1.** Investigation of the relationship between nymph phenotype, cuticle pigment absorbance, and cuticle thickness during the transition from summer-form to winter-form in *C. chinensis*.

**Figure supplement 2.** Phylogenetic tree analysis of *CcBurs-α* and *CcBurs-β* with its homologs in other insect species.

**Figure supplement 3.** Spatio-temporal expression patterns of *CcBurs-α* and *CcBurs-β*.

**Figure supplement 4.** RNAi efficiency of *CcTRPM* after dsRNA treatment at 3, 6, and 10 days by qRT-PCR under 10 °C condition (n=3).

those after *CcTRPM* knockdown (***Supplementary file 1b***). Additionally, dsCcBurs-α feeding (25.48%), dsCcBurs-β feeding (26.03%), or both feeding (11.84%) obviously decreased the transition percent from summer-form to winter-form compared to dsEGFP feeding (84.02%) (***Figure 2H–I***). Together, these data suggest that the two subunits of Bursicon, *CcBurs-α,* and *CcBurs-β*, are essential for the transition from summer-form to winter-form of *C. chinensis* by affecting cuticle contents and thickness.

### *CcBurs-R* was identified as the Bursicon receptor in *C. chinensis*

To study the role of neuropeptide Bursicon in seasonal polyphenism, we identified a leucine-rich repeat-containing G protein-coupled receptor and named it as Bursicon receptor *CcBurs-R* (GenBank: OR488626). The open reading frame of *CcBurs-R* is 3498 bp long and encodes an 1165 amino acid protein with a predicted molecular weight of 118.61 kDa, a theoretical *pI* of 8.64, and seven predicted transmembrane domains. Multiple alignment analysis showed a high degree of conservation in the transmembrane domains of *CcBurs-R* with *Burs-R* sequences from other four selected insect species (***Figure 3A***). Phylogenetic tree analysis indicated that *CcBurs-R* is most closely related to the *DcBurs-R* homologue (*D. citri*, KAI5703609.1) in an evolutionary relationship, and both are important Hemiptera pest of fruit trees (***Figure 3B***). The potential tertiary protein structure of *CcBurs-R* and its molecular docking with *CcBurs-α* and *CcBurs-β* were constructed using the online server Phyre2 and modified with PyMOL-v1.3r1 software (***Figure 3C***).

Development expression pattern indicated that *CcBurs-R* had relatively lower levels of expression in eggs and nymphs, but extremely higher levels in adult stages (***Figure 3—figure supplement 1A–B***). The mRNA level of *CcBurs-R* was higher in each developmental stage of the winter-form than summer-form, suggesting its important role in the transition from summer-form to the winter-form. In terms of tissue-specific expression, *CcBurs-R* was found to be present in all determined tissues, with relatively higher expression in five tissues (head, cuticle, midgut, wings, and foot) of winter-form than summer-form (***Figure 3—figure supplement 1C***). To confirm that *CcBurs-R* is the Bursicon receptor of *C. chinensis*, we employed a fluorescence-based assay to quantify calcium ion concentrations and investigate the binding affinities of bursicon heterodimers and homodimers to the Bursicon receptor across varying concentrations. Our findings suggest that activation of the receptor by the burs α-β heterodimer leads to significant alterations in intracellular calcium ion levels, whereas stimulation with burs α-α and burs β-β homodimers, in conjunction with Adipokinetic hormone (AKH), maintains consistent intracellular calcium ion levels. Consequently, this research definitively identifies *CcBurs-R* as the Bursicon receptor (***Figure 3—figure supplement 2***). In addition, we determined the effect of *CcBurs-α*

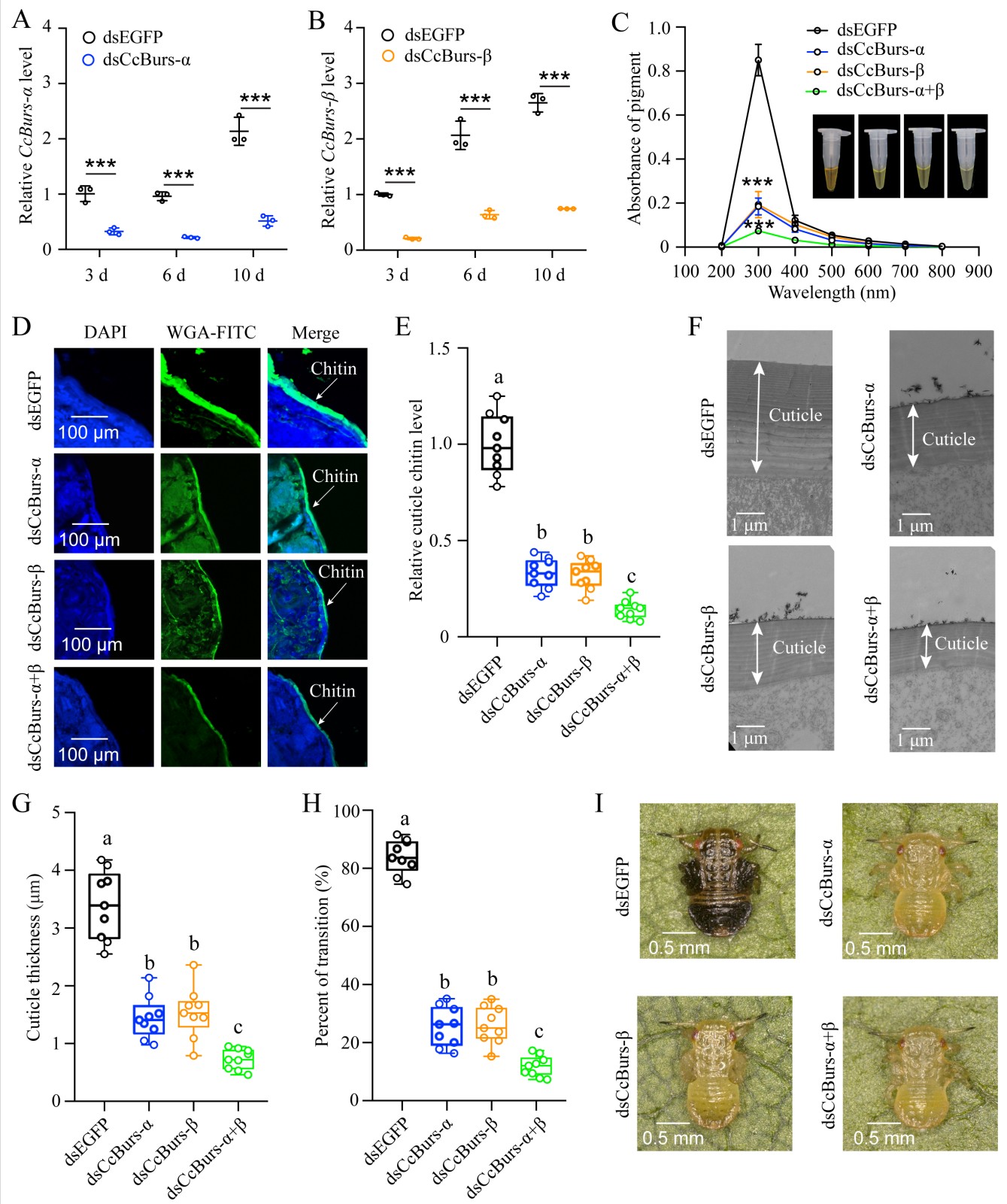

**Figure 2.** Neuropeptide Bursicon was essential for the transition from summer-form to winter-form in *C. chinensis*. (**A-B**) RNAi efficiency of *CcBurs*-α and *CcBurs*-β after dsRNA treatment at 3, 6, and 10 days by qRT-PCR under 10 °C condition (n=3). (**C-I**) Effect of RNAi-mediated knockdown of *CcBurs*-α and *CcBurs*-β on absorbance of total cuticle pigment, relative cuticle chitin content, cuticle thickness of the thorax, transition percent, and phenotypic changes of first instar nymphs compared to dsEGFP treatments (n=9). Data in 2 A and 2B are shown as the mean ± SE with three independent biological

*Figure 2 continued on next page*

*Figure 2 continued*

replications, with at least 50 nymphs for each replication. Data in 2 C, 2E, and 2 G are presented as mean ± SE with three biological replications, with three technical replications for each biological replication. Data in 2 H are presented as mean ± SE with nine biological replications. Statistically significant differences were determined using pair-wise Student's *t*-test, and significance levels were denoted by ***p<0.001. Different letters above the bars indicate statistically significant differences (p<0.05), as determined by ANOVA followed by a Turkey's HSD multiple comparison test in SPSS 26.0 software.

or *CcBurs-β* knockdown on its mRNA expression. qRT-PCR results showed that RNAi-mediated knockdown of *CcBurs-α* or *CcBurs-β* significantly decreased *CcBurs-R* expression after dsRNA feeding at 3, 6, and 10 days compared to the dsEGFP group under 10 °C condition (*Figure 3D–E*). Moreover, the heterodimer protein of *CcBurs-α+β* fully rescued the effect of RNAi-mediated knockdown on *CcBurs-R* expression, while α+α or β+β homodimers did not (*Figure 3F*). Additional results demonstrated that 10 °C treatment markedly increased *CcBurs-R* expression compared to 25 °C treatment, and *CcTRPM* knockdown obviously decreased the mRNA level of *CcBurs-R* compared to the dsEGFP treatment (*Figure 3G–H*). Therefore, these findings indicate that *CcBurs-R* is the Bursicon receptor of *C. chinensis* and is regulated by a low temperature of 10 °C and *CcTRPM*.

## *CcBurs-R* regulated the transition from summer-form to winter-form

The function of *CcBurs-R* in seasonal polyphenism was further investigated using RNAi technology. qRT-PCR results revealed that RNAi efficiency was 66–82% after dsCcBurs-R feeding for 3, 6, and 10 days compared to the dsEGFP treatments (*Figure 4A*). As shown in *Figure 4B–F*, the total pigment extraction at a wavelength of 300 nm (0.14 *vs* 0.85), cuticle chitin content (0.31 *vs* 1.00), and cuticle thicknesses (1.34 μm *vs* 3.39 μm) were all significantly decreased in dsCcBurs-R-treated nymphs compared to the dsEGFP control. Expectedly, the results of pigmentation absorbance and cuticle thickness after *CcBurs-R* knockdown were similar to those of *CcTRPM*, *CcBurs-α*, or *CcBurs-β* knockdown (*Supplementary file 1b*). In addition, RNAi-mediated down-regulation of *CcBurs-R* expression markedly affected the transition percent from summer-form to winter-form compared to dsEGFP feeding (26.70% *vs* 83.79%), while feeding the heterodimer protein of *CcBurs-α+β* (200 ng/μL) could fully rescue the effect of *CcBurs-R* knockdown on the transition percent (*Figure 4G–H*). Therefore, our results suggest that *CcBurs-R* mediates the transition from summer-form to winter-form by directly affecting cuticle contents and thickness.

Since *CcTre1* and *CcCHS1*, two rate-limiting enzyme genes in the chitin biosynthesis pathway, have been demonstrated to be involved down-stream in this transition of *C. chinensis*, we next investigated the relationship between Bursicon signal and these two genes. The results showed that the mRNA levels of *CcTre1* and *CcCHS1* were obviously decreased in dsCcBurs-α, dsCcBurs-β, or dsCcBurs-R feeding nymphs on the sixth day compared to the control (*Figure 4I–J*) This data indicates that *CcBurs-R* functions up-stream of the chitin biosynthesis pathway and is involved in the transition from summer-form to winter-form in *C. chinensis*.

## miR-6012 directly targeted *CcBurs-R* by inhibiting its expression

To determine if miRNAs are involved in the regulation of the Bursicon hormone in the seasonal polyphenism of *C. chinensis*, we amplified the 3'UTR of *CcBurs-R* and predicted relevant miRNAs. Four miRNAs, including miR-6012, miR-375, miR-2796, and miR-1175, were predicted to have binding sites in the 3'UTR of *CcBurs-R* by two software programs, miRanda and Targetscan (*Figure 5A*). To confirm the target relationship, in vitro dual-luciferase reporter assays were performed. After introducing the 3'UTR full sequence of *CcBurs-R* into the pmirGLO vector, the relative luciferase activity was significantly reduced compared to the negative control in the presence of agomir-6012, while there was no change with the other three miRNAs (*Figure 5B*). Next, in vivo, RNA immunoprecipitation results showed that the expression levels of *CcBurs-R* and miR-6012 increased approximately 15-fold and 23-fold, respectively, in the Ago-1 antibody-mediated RNA complex of agomir-6012 fed nymphs compared to the IgG control (*Figure 5C* and *Figure 5—figure supplement 1A–B*). FISH results indicated that *CcBurs-R* and miR-6012 had opposite expression trends during the developmental stages and were co-expressed in the third instar nymphs (*Figure 5D*). Co-localization implies direct interaction between miR-6012 and *CcBurs-R*, while the opposite expression pattern suggests that miRNAs have inhibitory effects on target genes. qRT-PCR results also revealed that low temperature prompted

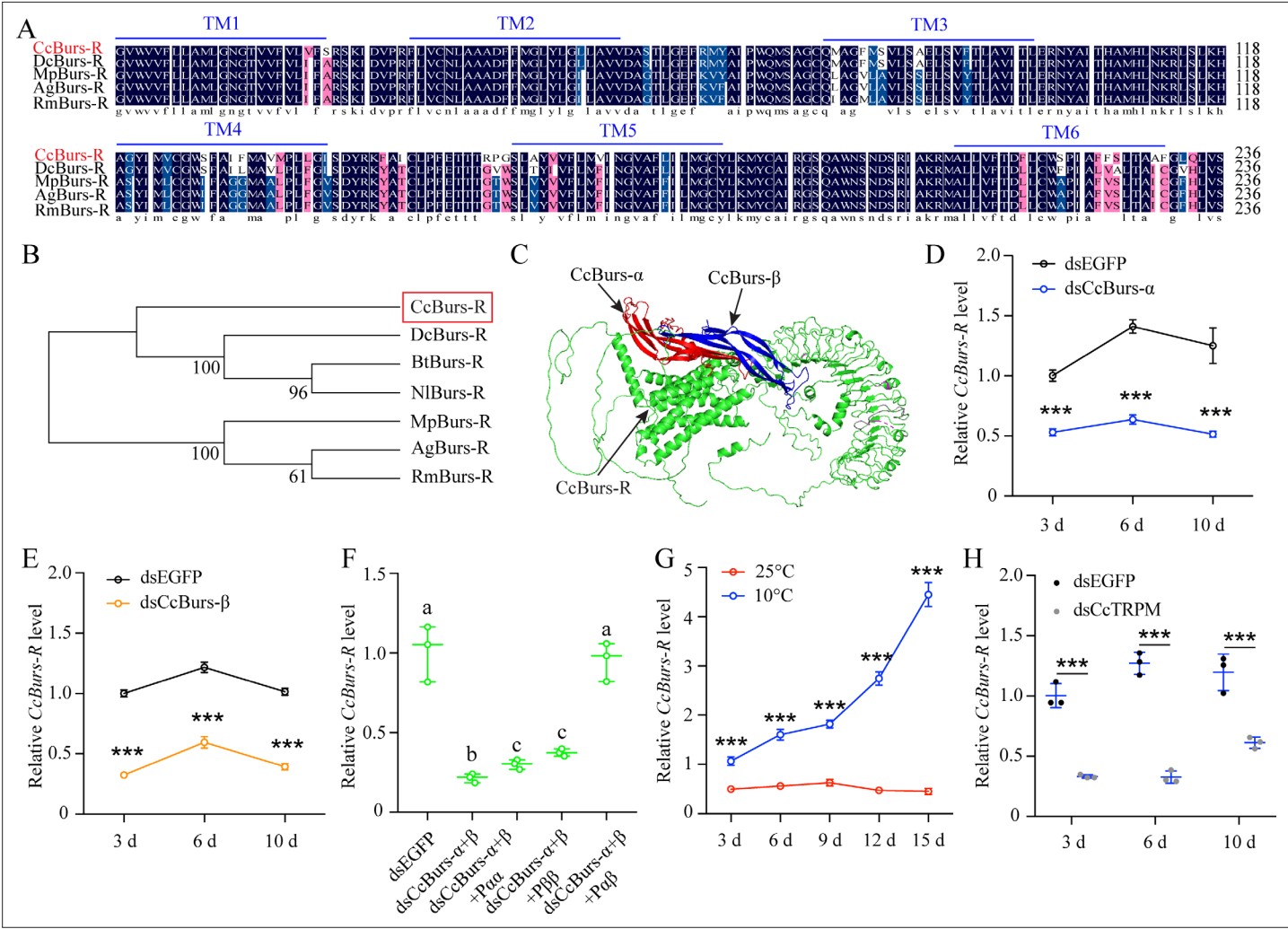

**Figure 3.** *CcBurs-R* was identified as the Bursicon receptor in *C. chinensis*. (**A**) Multiple alignments of the amino acid sequences of the *CcBurs-R* transmembrane domain with homologs from four other insect species. The transmembrane domain from TM1 to TM6 is indicated by blue horizontal lines. *CcBurs-R* (*C. chinensis*, OR488626), *DcBurs-R* (*D. citri*, KAI5703609.1), *MpBurs-R* (*M. persicae*, XP_022172830.1), *AgBurs-R* (*Aphis gossypii*, XP_027844917.2), *RmBurs-R* (*Rhopalosiphum maidis*, XP_026817427.1). The corresponding GenBank accession number is as follows. (**B**) Phylogenetic tree analysis of *CcBurs-R* with its homologs in six other insect species. *BtBurs-R* (*Bemisia tabaci*, XP_018898471.1), *NlBurs-R* (*N. lugens*, XP_022198758.2). (**C**) Predicted protein tertiary structure of *CcBurs-R* and its binding with *CcBurs-α* and *CcBurs-β*. (**D-E**) Effect of *CcBurs-α* and *CcBurs-β* knockdown on the mRNA expression of *CcBurs-R* at 3, 6, and 10 d, respectively (n=3). (**F**) CcBurs-α+β heterodimer protein could rescue the *CcBurs-R* expression after knockdown of *CcBurs-α* and *CcBurs-β* together. (**G**) Relative mRNA expression of *CcBurs-R* after 25 °C or 10 °C treatment at 3, 6, 9, 12, and 15 days (n=3). (**H**) Effect of temperature receptor *CcTRPM* knockdown on the mRNA expression of *CcBurs-R* at 3, 6, and 10 days (n=3). Data in 3D-3H are shown as the mean ± SE with three independent biological replications, with at least 50 nymphs for each replication. Statistically significant differences were determined using pair-wise Student's *t*-test in SPSS 26.0 software, and significance levels were denoted by ***$p<0.001$. Different letters above the bars indicated statistically significant differences ($p<0.05$), as determined by ANOVA followed by a Turkey's HSD multiple comparison test in SPSS 26.0 software.

The online version of this article includes the following figure supplement(s) for figure 3:

**Figure supplement 1.** Spatio-temporal expression patterns of *CcBurs-R* in both summer-form and winter-form by qRT-PCR (n=3).

**Figure supplement 2.** Concentration-response relationships for peptides tested on *Ccburs-R*.

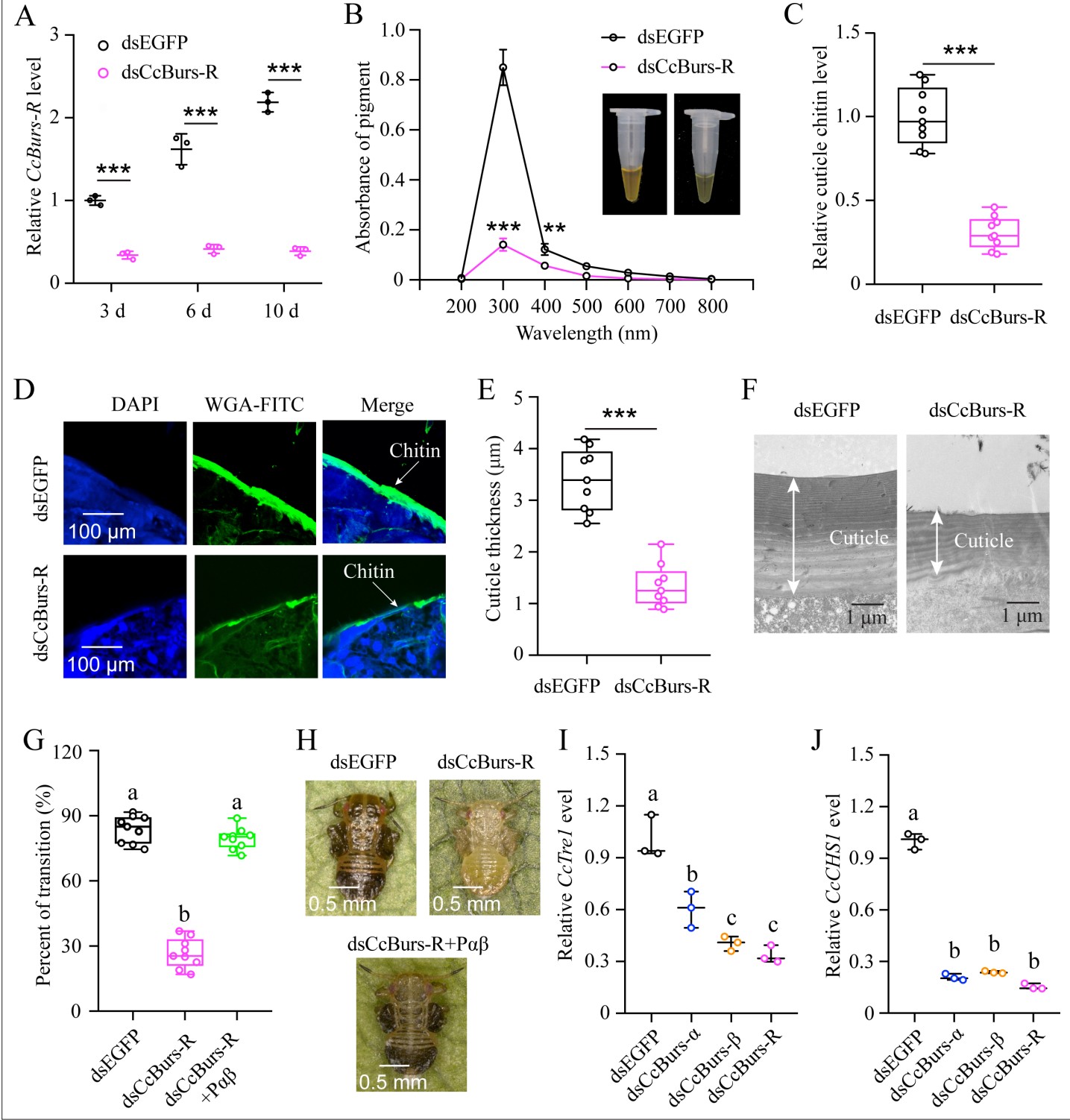

**Figure 4.** *CcBurs-R* directly mediated the transition from summer-form to winter-form in *C. chinensis*. (**A**) RNAi efficiency of *CcBurs-R* after dsRNA treatment at 3, 6, and 10 days by qRT-PCR under 10 °C condition (n=3). (**B-H**) Effect of RNAi-mediated knockdown of *CcBurs-R* on the absorbance of total cuticle pigment, relative cuticle chitin content, cuticle thickness of the thorax, transition percent, and phenotypic changes of first instar nymphs compared to dsEGFP treatments under 10 °C condition (n=9). (**I-J**) Relative mRNA expression of *CcTre1* and *CcCHS1* afterknockdown of *CcBurs-α,CcBurs-β,* and *CcBurs-R* at 10 d, separately (n=3). Data in 4 A, 4I, and 4 J are shown as the mean ± SE with three independent biological replications, with at least 50 nymphs for each replication. Data in 4B, 4 C, and 4E are presented as mean ± SE with three biological replications, with three technical replications for each biological replication. Data in 4 G are presented as mean ± SE with nine biological replications. Statistically significant differences

*Figure 4 continued on next page*

*Figure 4 continued*

were determined using pair-wise Student's *t*-test, and significance levels were denoted by **p<0.01 and ***p<0.001. Different letters above the bars indicate statistically significant differences (p<0.05), as determined by ANOVA followed by a Turkey's HSD multiple comparison test in SPSS 26.0 software.

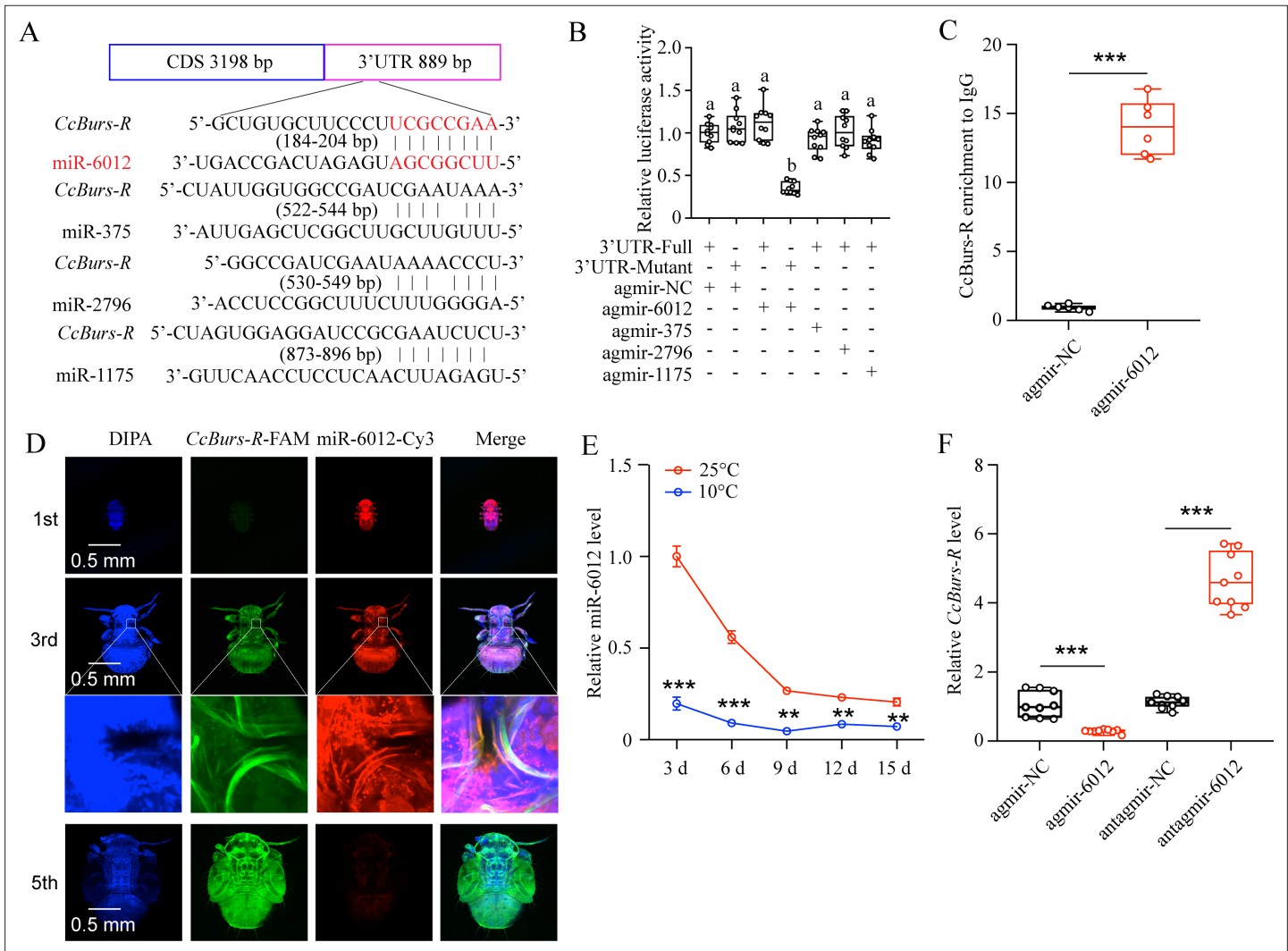

**Figure 5.** miR-6012 directly targeted *CcBurs-R* to inhibit its expression. (**A**) Predicted binding sites of four miRNAs in the 3'UTR of *CcBurs-R*. (**B**) In vitro confirmation of the target relationship between miR-6012 and *CcBurs-R* using dual luciferase reporter assays. (**C**) In vivo validation of miR-6012 directly targeting *CcBurs-R* using RNA-binding protein immunoprecipitation (RIP) assay. (**D**) Co-localization of miR-6012 and *CcBurs-R* in different development stages of *C. chinensis* using FISH. (**E**) Effect of different temperature treatments on the expression of miR-6012 by qRT-PCR. (**F**) Effect of miR-6012 agomir and antagomir treatments on the mRNA level of *CcBurs-R* at 6 days under 10 °C conditions. Data in 5B and 5 F are presented as the mean ± SE with nine biological replicates. Results of 5 C and 5E are indicated as the mean ± SE with six or three biological replicates. Statistically significant differences were determined using pair-wise Student's *t*-test, and significance levels were denoted by **p<0.01 and ***p<0.001. Different letters above the bars represent statistically significant differences (p<0.05), as determined by ANOVA followed by a Turkey's HSD multiple comparison test in SPSS 26.0 software.

The online version of this article includes the following figure supplement(s) for figure 5:

**Figure supplement 1.** Enrichment of miR-6012 by antibody against Ago1 in agomir-6012 treated group compared with agomir-NC group.

the expression of *CcBurs-R*, while miR-6012 had an inhibitory effect (*Figure 5E–F*). These data suggest that miR-6012 directly targets *CcBurs-R* by inhibiting its expression.

## miR-6012 mediated the seasonal polyphenism of *C. chinensis* by targeting *CcBurs-R*

To decipher the function of miR-6012 in regulating seasonal polyphenism, we increased its abundance by feeding agomir-6012 to the first instar nymphs. qRT-PCR results indicated that the expression levels of miR-6012 were markedly higher at 3, 6, and 10 days after agomir-6012 feeding compared to the agomir-NC control (*Figure 6A*). Furthermore, the results showed that agomir-6012 treatments significantly affected pigmentation absorbance, cuticle chitin content, cuticle thicknesses, the transition percent from summer-form to winter-form, and morphological phenotype compared to the negative control of agomir-NC feeding (*Figure 6B–H*). Additionally, agomir-6012 feeding also inhibited the mRNA expression of *CcTre1* and *CcCHS1* (*Figure 6I*). Together, these results display that miR-6012 plays an important role in the transition from summer-form to winter-form in *C. chinensis*.

## Discussion

Polyphenism is a conserved adaptive mechanism in species ranging from insects to mammalian, and evidence is mounting that it also extends to many nematode and fish species (*Stockton et al., 2018*; *Yang and Andrew Pospisilik, 2019*). Seasonal polyphenism can provide overwintering species with better adaptability to extreme climates through beneficial shifts in morphology, physiology, or behavior (*Simpson et al., 2011*). Physiological studies have shown that the neuroendocrine hormone system communicates environmental signals to facilitate downstream morphology and physiology transformation (*Zera and Denno, 1997*; *Overgaard and MacMillan, 2017*). Having a good model is extremely important for answering specific scientific questions (*Bhardwaj et al., 2020*). In *C. chinensis*, cuticle pigment absorbance and cuticle thickness, both have an increasing trend over time under 10 °C conditions, and showed very high correlation with the nymph phenotype of cuticle tanning during the transition from summer-form to winter-form (*Zhang et al., 2023*; *Figure 1—figure supplement 1*). To clarify the role of neuropeptide Bursicon in the seasonal polyphenism of *C. chinensis*, we identified two Bursicon subunits, *CcBurs-α* and *CcBurs-β*, in this study. The SDS-PAGE results of non-reduced and reduced gels showed that *CcBurs-α* and *CcBurs-β* can form both homodimers (α+α or β+β) and a heterodimer (α+β) (*Figure 1D*). During the transition of the *C. chinensis* between two forms, this study focused on the overall phenotypic changes. Therefore, for qPCR experiments, whole *C. chinensis* samples were selected for analysis. Temporal expression patterns showed that *CcBurs-α* and *CcBurs-β* have very similar gradually increasing expression trends and higher expression in winter-form than summer-form (*Figure 1—figure supplement 3C–D*), indicating that Bursicon may play a significant role in winter-form. This result is consistent with the report on gypsy moths, where transcript levels of *Ldbursicon* in adult stages were higher than in larvae (*Zhang et al., 2022b*). The transcript levels of both subunits were higher in the head and cuticle of winter-form compared to summer-form, implying a potential role of Bursicon in seasonal polyphenism of *C. chinensis* (*Figure 1—figure supplement 3E–F*; *Luan et al., 2006*).

As the transition of *C. chinensis* from summer-form to winter-form is regulated by a low temperature of 10 °C and *CcTRPM*, we next determined the effect of 10 °C treatment and *CcTRPM* RNAi on the expression of *CcBurs-α* and *CcBurs-β*. As expected, 10 °C treatment significantly increased the expression of *CcBurs-α* and *CcBurs-β*, while *CcTRPM* RNAi markedly decreased their mRNA levels (*Figure 1E–H*). This is the first report on the relationship between the neuropeptide Bursicon and low temperature. Further results from RNAi-mediated knockdown of *CcBurs-α*, *CcBurs-β*, or both showed that Bursicon prominently regulates the transition from summer-form to winter-form in *C. chinensis* by affecting cuticle pigment content, cuticle chitin content, and cuticle thickness (*Figure 2C–I*). Moreover, the presence of both thin and thick chitin layers observed in the dsEGFP treatment of *Figure 2D* could potentially be ascribed to the chitin content in the insect midgut or fat body as previously discussed (*Zhu et al., 2016*). It is notable that during the process of cuticle staining, the chitin located in the midgut and fat body of *C. chinensis* may exhibit green fluorescence, leading to the appearance of a thin chitin layer. In many insects, such as *Drosophila* and *T. castaneum*, Bursicon is believed to be the main hormone responsible for cuticle tanning (*Luo et al., 2005*; *Bai and Palli, 2010*). However,

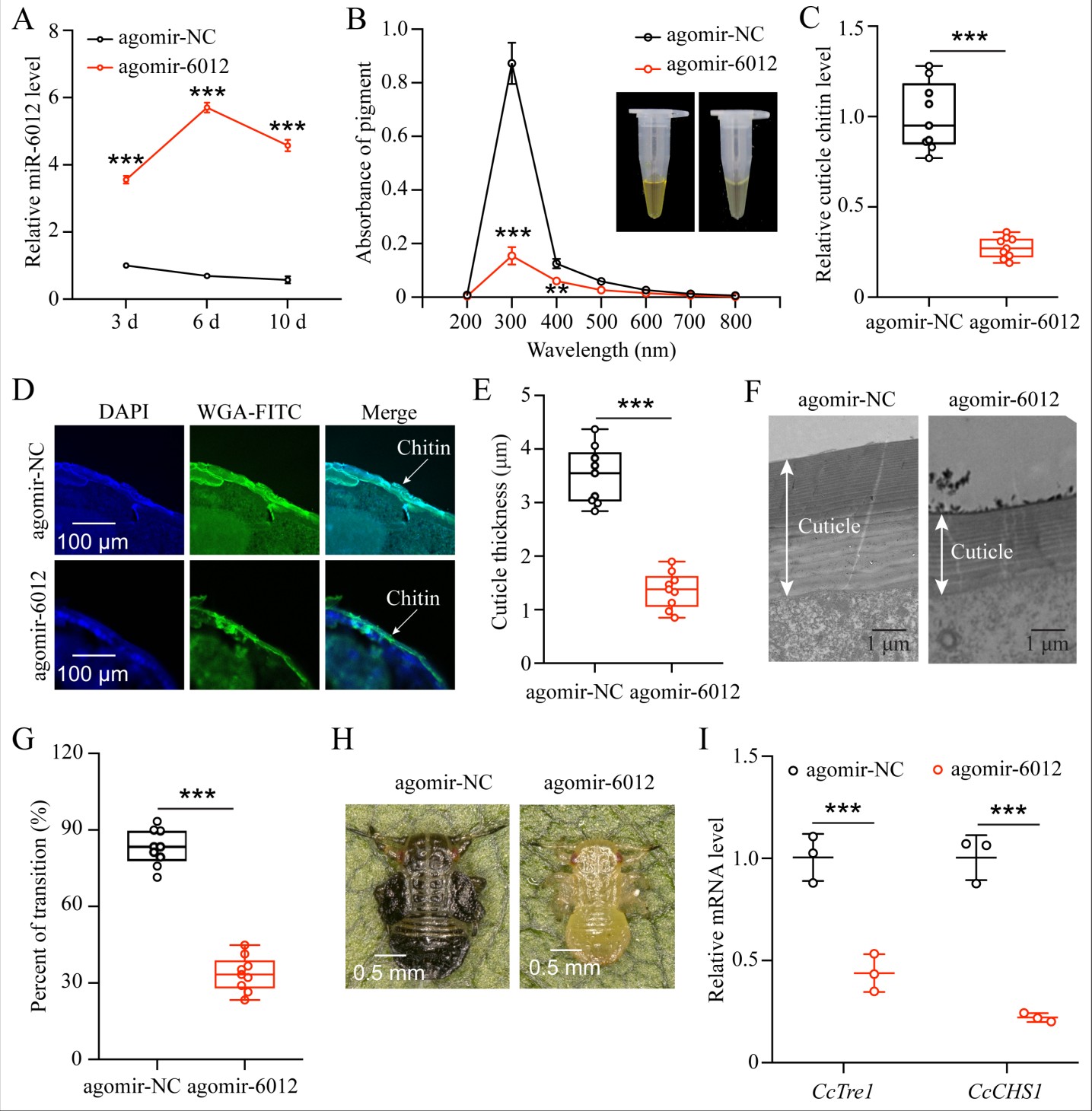

**Figure 6.** miR-6012 targeted *CcBurs-R* to mediate the seasonal polyphenism in *C. chinensis*. (**A**) Expression of miR-6012 after agomir-6012 treatment at 3, 6, and 10 days by qRT-PCR under 10 °C condition (n=3). (**B-H**) Effect of agomir-6012 treatment on absorbance of total cuticle pigment, relative cuticle chitin content, cuticle thickness of the thorax, transition percent, and phenotypic changes of first instar nymphs compared to agomir-NC treatments under 10 °C condition (n=9). (**I**) Relative mRNA expression of *CcTre1* and *CcCHS1* after agomir-6012 treatment at 6 days, separately (n=3). Data in 6 A and 6I are shown as the mean ± SE with three independent biological replications, with at least 50 nymphs for each replication. Data in 6 C and 6E are presented as mean ± SE with three biological replications of three technical replications for each biological replication. Data in 6B and 6 G are presented as mean ± SE with nine biological replications. Statistically significant differences were determined using pair-wise Student's *t*-test, and significance levels were denoted by **p<0.01 and ***p<0.001.

the knockdown of Bursicon subunits did not cause visible defects in cuticle sclerotisation or pigmentation of *Bombyx mori* and *Lymantria dispar* adults (*Huang et al., 2007*; *Zhang et al., 2022b*). The insect cuticle typically comprises three distinct layers (endocuticle, exocuticle, and epicuticle), with the thickness of each layer varying among different insect species. Cuticle differentiation is closely linked to the molting cycle of insects (*Mrak et al., 2017*). In our study, nymphal cuticles exhibited normal differentiation patterns, characterized by a thin epicuticle and comparable widths of the endocuticle and exocuticle following dsEGFP treatment, as illustrated in *Figures 2F and 4F*. Conversely, nymphs treated with dsCcBurs-α, dsCcBurs-β, and dsCcburs-R displayed impaired development, manifesting only the exocuticle without a discernible endocuticle layer. These findings suggest that bursicon genes and their receptor play a pivotal role in regulating insect cuticle development (*Costa et al., 2016*). These researches indicate that Bursicon may not be necessary for cuticle tanning in all insects. Although the reactions involved in cuticle tanning are well-known, further studies are needed to understand how Bursicon mediates the seasonal polyphenism of *C. chinensis*.

To further elucidate the role of the Bursicon signal in seasonal polyphenism, we identified the Bursicon receptor of *CcBurs-R* in *C. chinensis*. Temporal and spatial expression patterns of *CcBurs-R* were very similar to those of *CcBurs-α* and *CcBurs-β*, and it also had higher expression in winter-form than summer-form (*Figure 3—figure supplement 1*). By comparing its expression profiles with those in other insects, we can conclude that the spatio-temporal expression of the Bursicon receptor is related to the specificity of insect species. The activation of *CcBurs-R* by the burs α-β heterodimer exhibited a robust dose-dependent pattern. Conversely, no activation was observed when *CcBurs-R* was transfected with an empty vector or exposed to burs α-α and burs β-β homodimers or AKH (*Figure 3—figure supplement 2*). In *D. melanogaster*, Bursicon comprises two cystine knot polypeptides, pburs and burs, which are known to stimulate a G protein-coupled receptor, *DLGR2* (*Luo et al., 2005*). Through a radioligand receptor assay, specific and high-affinity interactions between *C. chinensis* burs α-β heterodimer and *CcBurs-R* were successfully confirmed. A recent study indicated that silencing of *Burs-α*,*Burs-β*,or its receptor significantly affected the reproduction of *T. castaneum* (*Bai and Palli, 2010*). Knockdown of *CcBurs-α*,*CcBurs-β*, or both obviously decreased the expression of *CcBurs-R*, while feeding the heterodimer protein of α+β fully rescued *CcBurs-R* expression after knockdown of *CcBurs-α* and *CcBurs-β* together, which further confirmed the relationship between subunits and the receptor (*Figure 3D–F*). 10 °C treatment clearly improved the expression of *CcBurs-R*, but *CcTRPM* RNAi sharply reduced its mRNA level (*Figure 3G–H*). Notably, the elimination of *CcBurs-R* in *C. chinensis* obviously affected cuticle pigment content, cuticle chitin content, and cuticle thickness, leading to the failure of the transition from summer-form to winter-form (*Figure 4B–H*). Feeding the α+β heterodimer protein fully rescued the defect in the transition percent and morphological phenotype after *CcBurs-R* knockdown (*Figure 4G–H*). Following the administration of dsCcBur-R to *C. chinensis*, the expression of *CcBurs-R* exhibited a reduction of approximately 66–82% as depicted in *Figure 4A*, rather than complete suppression. Activation of endogenous *CcBurs-R* through feeding of the α+β heterodimer protein results in an increase in *CcBurs-R* expression, with the effectiveness of the rescue effect contingent upon the dosage of the α+β heterodimer protein. Consequently, the capacity of the α+β heterodimer protein to effectively mitigate the impacts of *CcBurs-R* knockdown on the conversion rate is clearly demonstrated. Therefore, these findings strongly support our hypothesis that Bursicon and its receptor are essential for the transition from summer-form to winter-form in *C. chinensis*. Actually, seasonal polyphenism is a complex process that may be regulated by multiple cascade reactions. Further studies are needed to clarify the regulatory mechanism of Bursicon and its receptor in mediating the seasonal polyphenism of *C. chinensis*.

In animals, miRNAs are essential for tissue development and behavioral evolution (*Lucas and Raikhel, 2013*). Previous studies have reported that many miRNAs function upstream of the neurohormone signaling pathway in insect polyphenism (*Suderman et al., 2006*). For example, miR-133 controls behavioral aggregation by targeting the dopamine synthesis gene in Locusts (*Yang et al., 2014*), and miR-9b targets insulin receptors to mediate dimorphism and wing development in aphids (*Shang et al., 2020*). In this study, we identified miR-6012 as a regulator of *CcBurs-R* in the Bursicon hormone signaling pathway for the first time. We found that miR-6012 was inhibited by a low temperature of 10 °C and targeted *CcBurs-R* by binding to its 3'UTR. When nymphs were treated with agomir-6012, they exhibited lower cuticle pigment content, reduced cuticle chitin content, and thinner cuticle thickness compared to the agomir-NC control under 10 °C condition. In addition,

agomir-6012 treatment markedly decreased the transition percent from summer-form to winter-form and affected the morphological phenotype compared to the control. The significant decreased in *CcTre1* and *CcCHS1* expression after agomir-6012 treatment suggested that miR-6012 also functions as the upstream regulator of chitin biosynthesis signaling.

In conclusion, our study uncovered a novel role of Bursicon and its receptor in regulating the seasonal polyphenism of *C. chinensis*, in addition to their known functions in cuticle-hardening of nymphs and wing expansion of adults. In *Figure 7*, we proposed a molecular working model to describe this novel mechanism. Under 10 °C conditions, the Bursicon signaling pathway is first activated in the head of *C. chinensis* by low temperature and *CcTRPM*. Then, *CcBurs-α* and *CcBurs-β* form a heterodimeric neuropeptide that acts on its receptor *CcBurs-R* to mediate the transition from summer-form to winter-form by affecting cuticle pigment content, cuticle chitin content, and cuticle thickness. Moreover, miR-6012 targets *CcBurs-R* to modulate the function of the Bursicon signaling pathway in this seasonal polyphenism. As a result, the first instar nymphs of summer-form develop into third instar nymphs of winter-form to better adapt to low-temperature adversity. Future research will focus on: (1) studying the combined effect of Bursicon with other neuro-hormones on the seasonal polyphenism of *C. chinensis*, (2) identifying the down-stream signaling of Bursicon in mediating this phenomenon through multi-omics and RNAi approaches.

## Materials and methods

### Insect rearing

*C. chinensis* populations of summer-form and winter-form were collected in June and December 2018, respectively, from pear orchards in Daxing, Beijing, China. The nymphs and adults of summer-form were reared on host plants in a greenhouse under conditions of 25 ± 1°C, a photoperiod of 12 L:12D, and a relative humidity of 65 ± 5% (*Zhang et al., 2023*). Meanwhile, the nymphs and adults of winter-form were reared at 10 ± 1°C with a photoperiod of 12 L:12D and a relative humidity of 25 ± 5% in an artificial incubator. Unless otherwise specified, the photoperiod of all subsequent treatments was 12 L: 12D. Korla fragrant pear seedlings, 2–3 years old with a height of 50–80 cm, were used as host plants and received conventional water and fertilizer management.

### Gene identification and sequence analysis

From the transcriptome database of *C. chinensis*, we obtained the predicted sequences of Bursicon subunits and its receptor. After sequencing validation, we named them *CcBurs-α* (GenBank accession number: OR488624), *CcBurs-β* (GenBank accession number: OR488625), and *CcBurs-R* (GenBank accession number: OR488626). The physicochemical properties of *CcBurs-α*, *CcBurs-β*, and *CcBurs-R* were analyzed using the online bioinformatics ProtParam tool (http://web.expasy.org/protparam/). The putative transmembrane domains of *CcBurs-R* were identified using the online software SMART (Simple Modular Architecture Research Tool). The tertiary protein structures of *CcBurs-α*, *CcBurs-β*, and *CcBurs-R* were predicted using the online server Phyre[2] (http://www.sbg.bio.ic.ac.uk/phyre2/html/page.cgi?id=index) and modified with PyMOL-v1.3r1 software. Homologous protein sequences from different insect species were searched using BLASTP in the NCBI database. Multiple alignments of the amino acid sequences for *CcBurs-α*, *CcBurs-β*, and *CcBurs-R* with other homologs were performed using DNAman software. Phylogenetic analysis was carried out based on the neighbor-joining (NJ) method in MEGA10.1.8 software.

### Bursicon protein expression and determination

To express the Bursicon proteins in HEK293T cells, we first inserted the ORF sequences of *CcBurs-α* or *CcBurs-β* into the modified vector pcDNA3.1-his-P2A-mCherry to construct the recombinant vectors of pcDNA3.1-CcBurs-α-his-P2A-mCherry and pcDNA3.1-CcBurs-β-his-P2A-mCherry using the *pEASY*-Basic Seamless Cloning and Assembly Kit (Cat# CU201, TransGen, Beijing, China) (*Supplementary file 1a*). After confirming the sequences through sequencing and obtaining endotoxin-free plasmids, the recombinant vectors of *CcBurs-α* or *CcBurs-β* were transfected into HEK293T cells either individually or simultaneously following the protocol of *TransIntro* EI Transfection Reagent (Cat# FT201, TransGen, Beijing, China). Control cells were transfected with the blank vector pcDNA3.1-his-P2A-mCherry (without *CcBurs-α* or *CcBurs-β* cDNA insert). After 6–10 hr of transfection, the serum-free DMEM cell

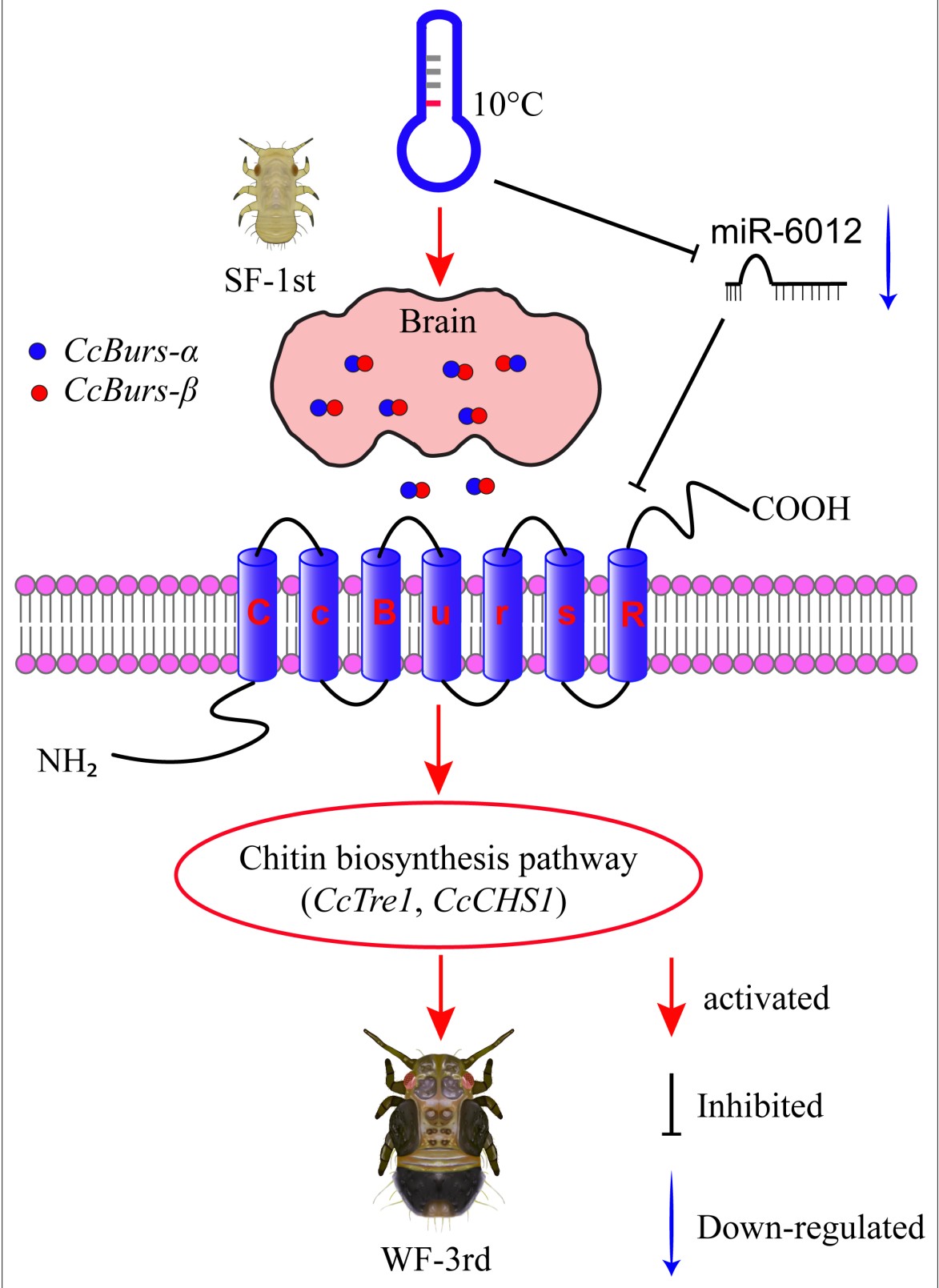

**Figure 7.** Schematic model of the novel functions of Bursicon signaling in the seasonal polyphenism of *C. chinensis* in response to low temperature. Under 10 °C conditions, low temperature significantly upregulated the expression of the Bursicon signaling pathway. *CcBurs-α* and *CcBurs-β* then formed a heterodimeric neuropeptide to activate their receptor *CcBurs-R*, which mediated the transition from summer-form to winter-form in *C. chinensis* by acting on the chitin biosynthesis pathway. Furthermore, low temperature inhibited the expression of miR-6012, relieving its inhibitory effects

*Figure 7 continued on next page*

on *CcBurs-R*. miR-6012 directly targeted *CcBurs-R*, contributing to the novel function of Bursicon signaling in seasonal polyphenism. Finally, the first instar nymphs of summer-form developed into third instar nymphs of winter-form in *C. chinensis.*

culture medium was replaced with fresh medium supplemented with 10% fetal bovine serum. After another 24 hr of incubation, the medium was replaced again with serum-free DMEM. The medium was collected and centrifuged at 1000×g for 10 min to remove cell debris after 48 h (*An et al., 2012a*). The expressed Bursicon proteins were purified using Ni-NTA His·bind resin (Cat# 70666, Merck, Germany). Then, western blotting was conducted to separate and identify these proteins using 15% SDS-PAGE (for reduced gel) and 12% SDS-PAGE (for non-reduced gel) with *ProteinFind* Anti-His Mouse Monoclonal Antibody (Cat# HT501, TransGen, Beijing, China). Lastly, the protein bands were imaged using enhanced chemiluminescence with the Azure C600 multifunctional molecular imaging system (USA).

## Heterologous expression and calcium mobilization assay

To construct the recombinant expression vector, the open reading frame sequence of *Ccburs-R* was integrated into a pcDNA3.1(+)-mCherry vector using the Vazyme ClonExpress II One Step Cloning Kit (Cat#C112, China) (homologous arm primers listed in *Supplementary file 1a*). The recombinant vector was prepared with the EndoFree Mid Plasmid Kit (catalog no. DP108, Tiangen, Beijing, China) and transfected into cultured cells in 96-well black plates or confocal dishes. The transiently transfected cells were cultured for 1–2 days in a 37 °C incubator, then stained with the Beyotime green fluorescent probe Fluo-4 AM (Cat# S1061, China) for approximately 30 min. Subsequently, $Ca^{2+}$ imaging and calcium concentration were assessed using Leica SP8 confocal microscopy (Wetzlar, Germany) and MD i3x microplate reader (San Jose, USA) following treatment with various dilutions of Bursicon protein.

## qRT-PCR for mRNA and miRNA

Samples for the temporal expression profile were collected at different developmental stages of summer form and winter form, including egg; nymphs of the first, second, third, fourth, and fifth instar; and adults of the first, third, and seventh day. For the tissue expression pattern, six types of tissue (head, cuticle, midgut, fat body, wings, and foot) were dissected from both summer-form and winter-form of fifth instar nymphs. To examine the effect of different temperature treatments on the expression of mRNAs and miRNAs, the newly hatched first instar nymphs of summer form were treated at 25°C and 10°C, respectively. Whole *C. chinensis* samples were collected at 3, 6, 9, 12, and 15 days after different temperature treatments. For the effect of *CcTRPM* knockdown on the transcription level of *CcBurs-α*,*CcBurs-β*, and *CcBurs-R* under 10 °C conditions, the newly hatched 1st instar nymphs of summer-form were fed with *CcTRPM* dsRNA, and the whole *C. chinensis* samples were collected on the third, sixth, and tenth day after dsRNA feeding. Each sample was performed in three replications, with approximately 100 individuals for each replication of egg samples and at least 50 insects were included for each nymph or adult sample. All samples were immediately stored at –80 °C for total RNA extraction.

Total RNAs were isolated from the above *C. chinensis* samples using TRNzol Universal (Cat# DP424, TIANGEN, Beijing, China) and miRcute miRNA isolation kit (Cat# DP501, TIANGEN, Beijing, China) for mRNA and miRNA, respectively, based on the manufacturer's protocol. The first-strand cDNA of mRNA or mature miRNA was synthesized from 500 ng or 1 µg of total RNAs using PrimeScript RT reagent kit with gDNA Eraser (Cat# RR047A, Takara, Kyoto, Japan) or miRcute Plus miRNA First-Strand cDNA Synthesis Kit (Cat# KR211, TIANGEN, Beijing, China) according to the instruction manual. The relative gene expression was quantified using TB Green *Premix Ex Taq* II (Tli RNaseH Plus) (Cat# RR820A, Takara, Kyoto, Japan) or miRcute Plus miRNA qPCR Detection Kit (Cat# FP411, TIANGEN, Beijing, China) in a total 20 µL reaction mixture on a CFX96 Connect Real-Time PCR System (Bio-Rad, Hercules, CA, USA). The conditions were as follows: denaturation for 3 min at 95 °C, followed by 40 cycles at 95 °C for 10 s, and then 60 °C for 30 s. *Ccβ-actin* (GenBank accession number: OQ658571) or U6 snRNA was used as the internal reference gene for qRT-PCR in *C. chinensis* (*Liu et al., 2020*; *Zhang et al., 2023*).To check for specificity, melting curves were analyzed for each data point (*Figure 1—figure supplement 3*, *Figure 3—figure supplement 1*,

*Figure 5—figure supplement 1*). The $2^{-\Delta\Delta CT}$ method (CT means the cycle threshold) was used to quantify gene expression of qRT-PCR data, where $\Delta\Delta CT$ is equal to $\Delta CT_{treated\ sample} - \Delta CT_{control}$ (*Livak and Schmittgen, 2001*).

## dsRNA synthesis and RNAi experiments

The synthesis of double-stranded RNA (dsRNA) and the stem-leaf device for dsRNA feeding was conducted as previously described (*Zhang et al., 2023*). Briefly, MEGAscript RNAi kit (AM1626, Ambion, California, USA) was used to synthesize dsRNA in vitro using primers ligated with T7 RNA polymerase promoter sequences at both ends (*Supplementary file 1a*). The dsRNAs were further purified with the phenol/chloroform method, air dried, dissolved in diethyl pyrocarbonate (DEPC)-treated nuclease-free water, and stored at –80 °C for later use. The purity and concentration of dsRNA were measured using ultraviolet spectrophotometry and gel electrophoresis.

For RNAi experiments, newly hatched first instar nymphs of summer-form were fed with dsRNAs (500 ng/μL) targeting different genes and then divided into three groups. (1) Whole *C. chinensis* samples were collected at 3, 6, and 10 d after dsRNAs feeding under 10 °C conditions for the RNAi efficiency analysis and gene expression analysis by qRT-PCR. (2) Whole *C. chinensis* nymph samples were collected at 12–15 days after dsRNA feeding for total cuticle pigment analysis, comparison of cuticle ultrastructure, cuticle chitin staining with WGA-FITC, and determination of cuticle chitin content under 10 °C condition using the following methods. (3) Morphological characteristics were observed every two days, and the number of summer-form and winter-form individuals was counted until the third instar under 10 °C condition, following the previous description (*Zhang et al., 2023*). For the rescue experiments, the dsRNA of *CcBurs-R* and proteins of burs α-α, burs β-β homodimers, or burs α-β heterodimer (200 ng/μL) were fed together.

## miRNA prediction and target validation with *CcBurs-R*

To study the post-translation function of *CcBurs-R*, the 3'UTR sequence of *CcBurs-R* was amplified using the specific primers (*Supplementary file 1a*) and the 3'-Full RACE Core Set with PrimeScript RTase kit (Cat# 6106, Takara, Kyoto, Japan). Two software programs, miRanda and Targetscan, were employed to predict miRNAs targeting *CcBurs-R*, following previously described methods (*Zhang et al., 2023*). The following methods were used to validate the target relationship between miRNAs and *CcBurs-R*.

### In vitro luciferase reporter gene assays

The full sequence of the 3'UTR or 3'UTR sequence with binding sites removed from *CcBurs-R* was amplified and inserted downstream of the luciferase gene in the pmirGLO vector (Promega, Wisconsin, USA) to construct recombinant plasmids. Agomir-6012 and antagomir-6012, chemically synthesized and modified RNA oligos with the same sequence or anti-sense oligonucleotides of miR-6012, were obtained from GenePharma (Shanghai, China). Agomir-NC and antagomir-NC, provided by the manufacturer, were used as negative controls. Approximate 500 ng of the recombinant plasmid and 275 nM of agomir were co-transfected into HEK293T cells using the Calcium Phosphate Cell Transfection Kit (Cat# C0508, Beyotime, Nanjing, China). After 24 hr of co-transfection, the activity of the luciferase enzymes was determined following the protocol of the Dual-Luciferase Reporter Assay System (Cat# E1910, Promega, Wisconsin, USA).

### In vivo RNA-binding protein immunoprecipitation assay (RIP)

The RIP assay was performed using theMagna RIP Kit (Cat# 17–704, Merck, Millipore, Germany) (*Zhang et al., 2023*). Fifty nymphs were collected after feeding with agomir-6012 or agomir-NC for 24 hr, and crushed with an auto homogenizer in ice-cold RIP lysis buffer. Magnetic beads were incubated with 5 μg of Ago-1 antibody (Merck, Millipore, Germany) or IgG antibody (Merck, Millipore, Germany) to form a magnetic bead-antibody complex. The target mRNAs were pulled down by the magnetic bead-antibody complex from the supernatants in the RIP lysates. The immunoprecipitated RNAs were released by digestion with protease K and quantification of *CcBurs-R* and miR-6012. Each experiment had six replicates.

### Fluorescence in situ hybridization (FISH)

The antisense nucleic acid probes for *CcBurs-R* (5'-GCGCUUGUGCUGCUUCUGCU-3') were labeled with FAM, and miR-6012 (5'-UGACCGACUAGAGUAGCGGCUU-3') was labeled with FITC (Gene-Pharma, Shanghai, China). In short, nymph samples at different stages were immersed in Carnoy's fixative for 24–48 hr at room temperature. After washing and decolorization, the samples were pre-hybridized three times using the hybridization buffer without the probes kept in the dark. For co-localization, two fluorescent probes (1 µM) were combined to hybridize the samples for about 12 hr in the dark. DAPI (1 µg/mL) was used to stain cell nuclei. The signals were observed and the images were recorded using a Leica SP8 confocal microscopy (Weztlar, Germany). To exclude false positives, RNAi-treated samples or no-probe samples were used as negative controls.

## Treatments of agomir-6012 and antagomir-6012

To study the temperature-dependent response threshold of miR-6012, the expression profiles of miR-6012 at various time points (3, 6, 9, 12, 15 days) subsequent to exposing *C. chinensis* to temperatures of 10°C and 25°C were measured. To examine the effect of miR-6012 on the mRNA expression of *CcBurs-R*,*CcTre1*, and *CcCHS1*, summer-form first instar nymphs were fed with agomir-6012 (1 µM) or antagomir-6012 (1 µM). Whole *C. chinensis* samples were first collected at 3, 6, and 10 days after feeding for agomir efficiency determination. Then, samples were collected at 6 days after treatment for total RNA extraction and qRT-PCR analysis. Agomir-NC and antagomir-NC were fed as a negative control.

To explore the function of miR-6012 in seasonal polyphenism, summer-form first instar nymphs were fed with agomir-6012 (1 µM) or agomir-NC (1 µM). Subsequently, cuticle ultrastructure comparison, cuticle chitin staining with WGA-FITC, determination of cuticle chitin content, and observation of morphological characteristics were performed as described in the following methods.

## Analysis of total cuticle pigment and cuticle chitin contents

To compare the difference in cuticle contents between summer-form and winter-form nymphs, the total cuticle pigment and cuticle chitin contents were determined. For the measurement of total cuticle pigment, the cuticle of dsRNA-treated nymphs was dissected and treated with acidified methanol (with 1% concentrated hydrochloric acid). The cuticle tissues were then ground and placed in a thermostatic oscillator at 200 rpm for 24 hr under 25 °C conditions. The total pigment extraction was obtained after filtering and centrifuging the supernatants through a 0.45 µm filter membrane. Pigments were not modified during extraction and the UV absorbance of the total pigment extraction at different wavelengths was determined using a NanoDrop 2000 (Thermo Fisher Scientific, USA) as previously described (*Futahashi et al., 2012*; *Osanai-Futahashi et al., 2012*).

For the analysis of cuticle chitin content, WGA-FITC staining was conducted as previously described (*Xie et al., 2022*; *Zhang et al., 2023*). Briefly, nymph samples were fixed with 4% paraformaldehyde and subjected to a gradient concentration dehydration with sucrose solution (10%, 20%, 30%). The dehydrated samples were then embedded in Tissue-Tek O.C.T. compound (Cat# 4583, SAKURA, Ningbo, China) after the pre-embedding stages at –25 °C. Ultra-thin sections (approximately 70 nm thickness) of the embedded material were cut using a Leica freezing ultra-cut microtome (CM1850, Leica, Weztlar, Germany). The sections were stained with WGA-FITC (50 µg/mL) and DAPI (10 µg/mL) for 15 min, followed by rinsing three times with sterile PBS buffer. Fluorescence images were acquired using a Leica SP8 confocal microscopy (Weztlar, Germany). To further quantify the cuticle chitin content, a chitin Elisa kit (Cat# YS80663B, Yaji Biotechnology, Shanghai, China) was used according to the previously described method (*Zhang et al., 2023*).

In order to determine the cuticle tanning threshold in *C. chinensis*, we examined the nymph phenotypes, cuticle pigment absorbance, and cuticle thickness levels in multiple time points (3, 6, 9, 12, 15 days) under two distinct temperatures of 10°C and 25°C. Each experimental condition encompassed nine independent biological replicates, with a minimum of 30 whole nymphs analyzed in each replicate for comprehensive assessment.

## Transmission electron microscopy assay

The TEM assay was performed as previously described (*Ge et al., 2019*; *Zhang et al., 2022a*; *Zhang et al., 2023*). In short, nymph samples without heads were fixed in 4% polyformaldehyde

(PFA) for 48 hr, followed by post-fixation in 1% osmium tetroxide for 1.5 hr. The samples were then dehydrated in a standard ethanol/acetone series, infiltrated, and embedded in a spurr medium. Subsequently, super thin sections (–70 nm) of the thorax were cut and stained with 5% uranyl acetate followed by Reynolds' lead citrate solution. The same dorsal region of the thorax was specifically chosen for subsequent fluorescence imaging or transmission electron microscopy assessments aimed at quantifying cuticle thickness. Lastly, the sections were observed, photographed, and measured using an HT7800 transmission electron microscope (Hitachi, Tokyo, Japan) operated at 120 kv. Regarding the measurement of cuticle thickness, use the built-in measuring ruler on the software to select the top and bottom of the same horizontal line on the cuticle. Measure the cuticle of each nymph at two close locations. Six nymphs were used for each sample. Randomly select 9 values and plot them.

## Statistical analysis

Figures preparation and statistical analysis were performed with GraphPad Prism 8.0 software and IBM SPSS Statistics 26.0, respectively. All data were shown as means ± SE (Standard Error of Mean) with different independent biological replications. Student's $t$-test was performed for pairwise comparisons to determine statistically significant differences between treatments and controls (*$p < 0.05$, **$p < 0.01$, and ***$p < 0.001$). One-way ANOVA followed by Tukey's HSD multiple comparison test was used for multiple comparisons in SPSS Statistics 26.0 (different letters denoted by $p < 0.05$).

## Acknowledgements

Thanks for the insect rearing by graduated students of Dongyue Zhang and Shili Meng from China Agricultural University. We appreciated transmission electron microscopy sample preparation from the microscopy laboratories of China Agricultural University. This work was funded by the National Natural Science Foundation of China (32202291) and China Agriculture Research System (CARS-28).

## Additional information

### Funding

| Funder | Grant reference number | Author |
| --- | --- | --- |
| National Natural Science Foundation of China | 32202291 | Songdou Zhang |
| China Agricultural Research System | CARS-28 | Xiaoxia Liu |

The funders had no role in study design, data collection and interpretation, or the decision to submit the work for publication.

### Author contributions

Zhixian Zhang, Conceptualization, Data curation, Software, Formal analysis, Validation, Investigation, Methodology, Writing – original draft; Jianying Li, Yilin Wang, Resources, Software, Formal analysis; Zhen Li, Data curation, Methodology; Xiaoxia Liu, Data curation, Formal analysis, Funding acquisition, Methodology; Songdou Zhang, Conceptualization, Data curation, Software, Formal analysis, Supervision, Funding acquisition, Validation, Investigation, Methodology, Writing – original draft, Project administration, Writing – review and editing

### Author ORCIDs

Zhixian Zhang ⓘ http://orcid.org/0000-0002-3898-3490
Songdou Zhang ⓘ https://orcid.org/0000-0002-3199-017X

Reviewer #1 (Public review): https://doi.org/10.7554/eLife.97298.3.sa1
Author response https://doi.org/10.7554/eLife.97298.3.sa2

# Additional files

## Supplementary files

• Supplementary file 1. The primers used in current study and comparison of pigmentation and cuticle thickness after genes knockdown. (**a**) List of primers used in this study. (b) Comparison of pigmentation and cuticle thickness after *CcTRPM*, *CcBurs-a*, *CcBurs-β*, and *CcBurs-R* knockdown.

• MDAR checklist

## Data availability

The published article includes all data generated or analyzed during this study. The full sequences of *CcBurs-α*, *CcBurs-β*, and *CcBurs-R* were submitted to GenBank database of NCBI (Accession number: OR488624, OR488625, and OR488626).

The following datasets were generated:

| Author(s) | Year | Dataset title | Dataset URL | Database and Identifier |
|---|---|---|---|---|
| Zhang Z, Li J, Wang Y, Li Z, Liu X, Zhang S | 2024 | *Cacopsylla chinensis* Bursicon alpha gene, complete sequence | https://www.ncbi. nlm.nih.gov/nuccore/ OR488624 | NCBI Nucleotide, OR488624 |
| Zhang Z, Li J, Wang Y, Li Z, Liu X, Zhang S | 2024 | *Cacopsylla chinensis* Bursicon beta gene, complete sequence | https://www.ncbi. nlm.nih.gov/nuccore/ OR488625.1 | NCBI Nucleotide, OR488625.1 |
| Zhang Z, Li J, Wang Y, Li Z, Liu X, Zhang S | 2024 | *Cacopsylla chinensis* Bursicon receptor gene, complete sequence | https://www.ncbi. nlm.nih.gov/nuccore/ OR488626.1 | NCBI Nucleotide, OR488626.1 |

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
