## [Editor Report · eLife Assessment]

This **important** study reports that the neurohormone, bursicon, and its receptor, play a role in the seasonal polyphenism of the bug Cacopsylla chinensis. Low temperature activates the bursicon signaling pathway during the transition from the summer to the winter form, affecting cuticle pigment and thickness as well as chitin content. The **solid** experiments reveal how bursicon signaling, which is modulated by the microRNA miR-6012, regulates features of polyphenism related to the exoskeleton, although it is less clear what the upstream regulatory events are.

---

## [Referee Report · Reviewer #1 (Public review)]

Bursicon is a key hormone regulating cuticle tanning in insects. While the molecular mechanisms of its function are rather well studied--especially in the model insect *Drosophila melanogaster*, its effects and functions in different tissues are less well understood. Here, the authors show that bursicon and its receptor play a role in regulating aspects of the seasonal polyphenism of Cacopsylla chinensis. They found that low temperature treatment activated the bursicon signaling pathway during the transition from summer form to winter form and affect cuticle pigment and chitin content, and cuticle thickness. In addition, the authors show that miR-6012 targets the bursicon receptor, CcBurs-R, thereby modulating the function of bursicon signaling pathway in the seasonal polyphenism of C. chinensis. This discovery expands our knowledge of the roles of neuropeptide bursicon action in arthropod biology.

Reviewer comments on revised version

(a) Major concerns

(1) The revision did not respond to the major concern regarding the threshold response that defines polyphenism. Therefore, it still falls short of the claims made, since the claims were not revised either. Specifically, the authors now include a time series of tanning at two different temperatures, demonstrating the time points at which the induced tanning proceeds (Fig. S1). However, the appropriate response to that comment would have temperatures on the x-axis, not time. Intermediate temperatures are needed to test whether the induction is a threshold response or simply a continuous norm of reaction.

(2) The authors also did not respond to the major comment regarding environmental induction of miR-6012 expression. Rather, Fig. 5E shows a time series under two temperatures, similar to the tanning time series. To test whether its induction is a threshold response (again, what defines polyphenism), a series of induction conditions is needed. Fig. 5E simply shows changes in expression over time under one induction temperature (25 ºC).

(3) Although the manuscript title has been changed, little to nothing else in the revised text addresses the concern that this study is about tanning in psyllids, not seasonal polyphenism. The other traits making up the polyphenism, as well as their threshold response, were not measured.

In summary, this revision failed to address most of the chief concerns of the review summary. This manuscript should be reframed as a study of tanning in a species other than *Drosophila*, and any claims about polyphenism (that is, an environmentally induced threshold trait) still need to be tested.

Regarding the other concerns raised by the reviewers:

(4) Issues related to the assignment of the receptor used as a bursicon receptor were satisfactorily addressed.

(5) Experiments regarding the timing of cuticle production presented in Supplementary Figure 1 are valuable, albeit, there are still some inaccuracies: (i) the layering of the cuticle is not given accurately as there are more than the 3 layers indicated in the manuscript; (ii), the reduced endocuticle in all relevant dsRNA cases suggests a massive molting defect that may underline the involvement of bursicon in molting in general, potentially masking its effect on morph transition. In other words, the phenotype is too strong to allow for the interpretation of its function with respect to morph transition. It would have been necessary to apply different concentrations of dsRNA in order to address this point. (iii) The developmental timing at 10oC vs. 25oC seem to be similar, although duration would be expected to be longer at 10oC; (iv) It would have been nice to see the days of development also for dsRNA injected animals.

(6) Another unresolved point regards the source and target tissue of bursicon signaling. Admittedly, this problem is difficult to solve in a small insect species.

---

## [Author Response]

The following is the authors’ response to the original reviews.

**Public Review:**
Summary:Bursicon is a key hormone regulating cuticle tanning in insects. While the molecular mechanisms of its function are rather well studied--especially in the model insect *Drosophila melanogaster*, its effects and functions in different tissues are less well understood. Here, the authors show that bursicon and its receptor play a role in regulating aspects of the seasonal polyphenism of *Cacopsylla chinensis*. They found that low temperature treatment activated the bursicon signaling pathway during the transition from summer form to winter form and affect cuticle pigment and chitin content, and cuticle thickness. In addition, the authors show that miR-6012 targets the bursicon receptor, *CcBurs-R*, thereby modulating the function of bursicon signaling pathway in the seasonal polyphenism of *C. chinensis*. This discovery expands our knowledge of the roles of neuropeptide bursicon action in arthropod biology.However, the study falls short of its claim that it reveals the molecular mechanisms of a seasonal polyphenism. While cuticle tanning is an important part of the pear psyllid polyphenism, it is not the equivalent of it. First, there are other traits that distinguish between the two morphs, such as ovarian diapause (Oldfield, 1970), and the role of bursicon signaling in regulating these aspects of polyphenism were not measured. Thus, the phenotype in pear psyllids, whereby knockdown bursicon reduces cuticle tanning seems to simply demonstrate the phenotypes of *Drosophila* mutants for bursicon receptor (Loveall and Deitcher, 2010, BMC Dev Biol) in another species (Fig. 2I, 4H). Second, the study fails to address the threshold nature of cuticular tanning in this species, although it is the threshold response (specifically, to temperature and photoperiod) that distinguishes this trait as a part of a polyphenism. Whereas miR-6012 was found to regulate bursicon expression, there no evidence is provided that this microRNA either responds to or initiates a threshold response to temperature. In principle, miR-6012 could regulate bursicon whether or not it is part of a polyphenism. Thus, the impact of this work would be significantly increased if it could distinguish between seasonal changes of the cuticle and a bona fide reflection of polyphenism.

Thanks for your valuable suggestion. We concur with the review’s comment that cuticle tanning does not equate to the *C. chinensis* polyphenism. To better reflect the core focus of our research, we have revised the title to "Neuropeptide Bursicon and its receptor mediated the transition from summer-form to winter-form of *Cacopsylla chinensis*".

In response to the reviewer's inquiry regarding the threshold nature of cuticular tanning in *C. chinensis*, we have included a detailed analysis of the phenotypic changes (including nymph phenotypes, cuticle pigment absorbance, and cuticle thickness) during the transition from summer-form to winter-form in *C. chinensis* at distinct time intervals (3, 6, 9, 12, 15 days) under different temperature conditions (10°C and 25°C). As shown in Figure S1, nymphs exhibit a light yellow and transparent coloration at 3, 6, and 9 days, while nymphs at 12 and 15 days display shades of yellow-green or blue-yellow under 25°C conditions. At 10°C conditions, the abdomen end turns black at 3, 6, and 9 days. By the 12 days, numerous light black stripes appear on the chest and abdomen of nymphs at 10°C. At 15 days, nymphs exhibit an overall black-brown appearance, featuring dark brown stripes on the left and right sides of each chest and abdominal section. Furthermore, the end of the abdomen and back display a large black-brown coloration at 10°C (Figure S1A). The UV absorbance of the total pigment extraction at a 300 nm wavelength markedly increases following 10°C exposure for 6, 9, 12, and 15 days compared to the 25°C treatment group (Figure S1B). Cuticle thicknesses also increased following 10°C exposure for 6, 9, 12, and 15 days compared to the 25°C treatment group (Figure S1C). The detailed results (L122-143), materials and methods (L647-652), and discussion (L319-322) have been added in our revised manuscript.

Regarding the response of miR-6012 to temperature, we have already determined its expression at 3, 6, 10 days under different temperatures in the previous Figure 5E. We now included additional time intervals (9, 12, 15 days) in the updated Figure 5E. Our results indicate a significant decrease in the expression levels of miR-6012 after 10°C treatment for 3, 6, 9, 12, 15 days compared to the 25°C treatment group. Detailed information regarding this has been integrated into the Materials and Methods (Line 608-610) of our revised manuscript.

Strengths:This study convincingly identifies homologs of the genes encoding the bursicon subunits and its receptor, showing an alignment with those of another psyllid as well as more distant species. It also demonstrates that the stage- and tissue-specific levels of bursicon follow the expected patterns, as informed by other insect models, thus validating the identity of these genes in this species. They provide strong evidence that the expression of bursicon and its receptor depend on temperature, thereby showing that this trait is regulated through both parts of the signaling mechanism.Several parallel measurements of the phenotype were performed to show the effects of this hormone, its receptor, and an upstream regulator (miR-6012), on cuticle deposition and pigmentation (if not polyphenism per se, as claimed). Specifically, chitin staining and TEM of the cuticle qualitatively show difference between controls and knockdowns, and this is supported by some statistical tests of quantitative measurements (although see comments below). Thus, this study provides strong evidence that bursicon and its receptor play an important role in cuticle deposition and pigmentation in this psyllid.The study identified four miRNAs which might affect bursicon due to sequence motifs. By manipulating levels of synthetic miRNA agonists, the study successfully identified one of them (miR-6012) to cause a cuticle phenotype. Moreover, this miRNA was localized (by FISH) to the cuticle, body-wide. To our knowledge, this is the first demonstrated function for this miRNA, and this study provides a good example of using a gene of known function as an entry point to discovering others influencing a trait. Thus, this finding reveals another level of regulation of cuticle formation in insects.Weaknesses:(1) The introduction to this manuscript does not accurately reflect progress in the field of mechanisms underlying polyphenism (e.g., line 60). There are several models for polyphenism that have been used to uncover molecular mechanisms in at least some detail, and this includes seasonal polyphenisms in Hemiptera. Therefore, the justification for this study cannot be predicated on a lack of knowledge, nor is the present study original or unique in this line of research (e.g., as reviewed by Zhang et al. 2019; DOI: 10.1146/annurev-ento-011118-112448). The authors are apparently aware of this, because they even provide other examples (lines 104-108); thus the introduction seems misleading as framed.

Thanks for your excellent suggestion. We have added the paper of Zhang et al. 2019 which recommended by reviewer (DOI: 10.1146/annurev-ento-011118-112448) in Line 57 of our revised manuscript. The statement has been revised to “However, the specific molecular mechanism underling temperature-dependent polyphenism still require further clarification” in Line 60-61 of our revised manuscript.

(2) The data in Figure 2H show "percent of transition." However, the images in 2I show insects with tanned cuticle (control) vs. those without (knockdown). Yet, based on the description of the Methods provided, there appears to be no distinction between "percent of transition" and "percent with tanning defects". This an important distinction to make if the authors are going to interpret cuticle defects as a defect in the polyphenism. Furthermore, there is no mention of intermediate phenotypes. The data in 2H are binned as either present or absent, and these are the phenotypes shown in 2I. Was the phenotype really an all-or-nothing response? Instead of binning, which masks any quantitative differences in the tanning phenotypes, the authors should objectively quantify the degree of tanning and plot that. This would show if and to what degree intermediate tanning phenotypes occurred, which would test how bursicon affects the threshold response. This comment also applies to the data in Figures 4G and 6G. Since cuticle tanning is present in more insect than just those with seasonal polyphenism, showing how this responds as a threshold is needed to make claims about polyphenism.

We appreciate your insightful comments. As shown in Figure 1 of our published paper (Zhang et al., 2013; doi.org/10.7554/eLife.88744.3) and Figure 2C-2I of the current manuscript, the transition from summer-form to winter-form entails not only external cuticular tanning but also alterations in internal cuticular chitin levels and cuticle thickness. While external cuticular tanning serves as a prominent and easily observable indicator of this transition, it is crucial to acknowledge that internal changes also play a significant role and should be taken into consideration. Therefore, we propose that the term "percent of transition" may be more suitable than "percent with tanning defects" to describe this process accurately.

In order to provide a more visually comprehensive understanding of the phenotypic changes during the transition from summer-form to winter-form, we have included images at different time points (3, 6, 9, 12, 15 days) under different temperature conditions in Figure S1A of our revised manuscript. Specifically, under the 10°C condition, nymphs exhibit abdomen tanning after 6 and 9 days of treatment, while the thorax remains untanned. By days 12 to 15, both the abdomen and thorax of the nymphs show tanning, resulting in the majority of summer-form nymphs transitioning into winter-form, as depicted in Figure 2I for comparison. This observation indicates the presence of a critical threshold for cuticle tanning of *C. chinensis* following exposure to 10°C. Nymphs that did not undergo the transition to winter-form succumbed to the cold, highlighting the absence of intermediate phenotypes at 12-15 days under the 10°C condition. The UV absorbance of the total pigment extraction at a 300 nm wavelength markedly increases following 10°C exposure for 6, 9, 12, and 15 days compared to the 25°C treatment group (Figure S1B). Additionally, cuticle thickness shows an increase following 10°C exposure for 6, 9, 12, and 15 days compared to the 25°C treatment group (Figure S1C). These results highlight the relationship between the threshold of cuticular tanning and the transition process. The detailed description and information have been added in Results (L122-143), Materials and Methods (L647-652), and Discussion (L319-322) of our manuscript.

(3) This study also does not test the threshold response of cuticle phenotypes to levels of bursicon, its receptor, or miR-6012. Hormone thresholds are the most widespread and, in most systems where polyphenism has been studied, the defining characteristic of a polyphenism (e.g., Nijhout, 2003, Evol Dev). Quantitative (not binned) measurements of a polyphenism marker (e.g., chitin) should be demonstrated to result as a threshold titer (or in the case of the receptor, expression level) to distinguish defects in polyphenism from those of its component trait.

Thanks for your valuable feedback. We have supplemented additional data on the phenotypes (Figure S1A), cuticle pigment absorbance (Figure S1B), cuticle thickness (Figure S1C), expression levels of bursicon (Figure 1E and 1F), its receptors (Figure 3G), and miR-6012 (Figure 5E) corresponding to nymphs treated over different time periods (3, 6, 9, 12, 15 days) under both 10°C and 25°C conditions in our revised manuscript.

While all these identified markers exhibit a strong correlation with the transition from summer-form to winter-form, it is important to note that they are not suitable as definitive thresholds due to the nature of relative gene expression quantification and chitin content assessment, rather than absolute quantitation. Further, given that tanning hormones are neuropeptides present in trace amounts in insects, unlike steroid hormones, determining their titers poses a considerable challenge.

(4) Cuticle issue:(a) Unlike Fig. 6D and F, Figs. 2D and F do not correspond to each other. Especially the lack and reduction of chitin in ds-a+b! By fluorescence microscopy there is hardly any signal, whereas by TEM there is a decent cuticle. Additionally, the dsGFP control cuticle in 2D is cut obliquely with a thick and a thin chitin layer. This is misleading.

Thanks for your insightful feedback. We have replaced the previous WGA chitin staining images in the dsCcbursα+β treatment of Figure 2D with new representative images aligning with Figure 2F. Furthermore, the presence of both thin and thick chitin layers observed in the dsEGFP treatment of Figure 2D could potentially be ascribed to the chitin content in the insect midgut or fat body as previously discussed (Zhu et al., 2016). It is notable that during the process of cuticle staining, the chitin located in the midgut and fat body of *C. chinensis* may exhibit green fluorescence, leading to the appearance of a thin chitin layer. A detailed analysis and elucidation of these observations have been added in the discussion section (Lines 347-352) of our revised manuscript.

Zhu KY, Merzendorfer H, Zhang W, Zhang J, Muthukrishnan S. Biosynthesis, Turnover, and Functions of Chitin in Insects. Annu Rev Entomol. 2016;61:177-196. doi:10.1146/annurev-ento-010715-023933.

(b) In Figs. 2F and 4F, the endocuticle appears to be missing, a portion of the procuticle that is produced post-molting. As tanning is also occurring post-molting, there seems to be a general problem with cuticle differentiation at this time point. This may be a timing issue. Please clarify.

Thank you for your suggestion. The insect cuticle typically comprises three distinct layers (endocuticle, exocuticle, and epicuticle), with the thickness of each layer varying among different insect species. Cuticle differentiation is closely linked to the molting cycle of insects (Mrak et al., 2017). In our study, nymphal cuticles exhibited normal differentiation patterns, characterized by a thin epicuticle and comparable widths of the endocuticle and exocuticle following dsEGFP treatment, as illustrated in Figure 2F and 4F. Conversely, nymphs treated with dsCcBurs-α, dsCcBurs-β, and dsCcburs-R displayed impaired development, manifesting only the exocuticle without a discernible endocuticle layer. These findings suggest that bursicon genes and their receptor play a pivotal role in regulating insect cuticle development (Costa et al., 2016). We have added some discussion about these results in Lines 356-367 of our revised manuscript.

Mrak, P., Bogataj, U., Štrus, J., & Žnidaršič, N. (2017). Cuticle morphogenesis in crustacean embryonic and postembryonic stages. Arthropod structure & development, 46(1), 77–95. https://doi.org/10.1016/j.asd.2016.11.001

Costa, C. P., Elias-Neto, M., Falcon, T., Dallacqua, R. P., Martins, J. R., & Bitondi, M. (2016). RNAi-mediated functional analysis of Bursicon genes related to adult cuticle formation and tanning in the Honeybee, *Apis mellifera*. PloS one, 11(12), e0167421. https://doi.org/10.1371/journal.pone.0167421

(c) To provide background information, it would be useful analyze cuticle formation in the summer and winter morphs of controls separately by light and electron microscopy. More baseline data on these two morphs is needed.

Thanks for your valuable feedback. To provide more background information about cuticle formation, we supplied the results of nymph phenotypes, cuticle pigment absorbance, and cuticle thickness at distinct time intervals (3, 6, 9, 12, 15 days) under different temperatures of 10°C and 25°C in Figure S1 of our revised manuscript. Hope these results can help better understand the baseline data on these two morphs.

(d) For the TEM study, it is not clear whether the same part of the insect's thorax is being sectioned each time, or if that matters. There is not an obvious difference in the number of cuticular layers, but only the relative widths of those layers, so it is difficult to know how comparable those images are. This raises two questions that the authors should clarify. First, is it possible that certain parts of the thoracic cuticle, such as those closer to the intersegmental membrane, are naturally thinner than other parts of the body? Second, is the tanning phenotype based on the thickness or on the number of chitin layers, or both? The data shown later in Figure 4I, J convincingly shows that the biosynthesis pathway for chitin is repressed, but any clarification of what this might mean for deposition of chitin would help to understand the phenotypes reported. Also, more details on how the data in Fig. 2G were collected would be helpful. This also goes for the data in Fig. 4 (bursicon receptor knockdowns).

Thanks for your great comment. The TEM investigation adhered to a standardized protocol was used as previous description (Zhang et al., 2023), Initially, insect heads were uniformly excised and then fixed in 4% paraformaldehyde. Subsequently, a consistent cutting and staining procedure was executed at a uniform distance above the insect's thorax. The dorsal region of the thorax was specifically chosen for subsequent fluorescence imaging or transmission electron microscopy assessments with the specific objective of quantifying cuticle thickness. Regarding the measurement of cuticle thickness, use the built-in measuring ruler on the software to select the top and bottom of the same horizontal line on the cuticle. Measure the cuticle of each nymph at two close locations. Six nymphs were used for each sample. Randomly select 9 values and plot them. The related description has been added in the Materials and Methods (Line 660-668) of our revised manuscript.

Zhang, S.D., Li, J.Y., Zhang, D.Y., Zhang, Z.X., Meng, S.L., Li, Z., & Liu, X.X. (2023). MiR-252 targeting temperature receptor *CcTRPM* to mediate the transition from summer-form to winter-form of *Cacopsylla chinensis*. *eLife*, *12*. https://doi.org/10.7554/eLife.88744

(5) Tissue issue:The timed experiments shown in all figures were done in whole animals. However, we know from *Drosophila* that Bursicon activity is complex in different tissues. There is, thus, the possibility, that the effects detected on different days in whole animals are misleading because different tissues--especially the brain and the epidermis, may respond differentially to the challenge and mask each other's responses. The animal is small, so the extraction from single tissue may be difficult. However, this important issue needs to be addressed.

Thanks for your excellent suggestion. We express our heartfelt appreciation to the reviewer for their valuable input regarding the challenges involved in dissecting various tissue sections from the diminutive early instar nymphs of *C. chinensis*. In light of the metamorphic transition of *C. chinensis* across developmental stages, this study concentrated on examining the extensive phenotypic alterations. Consequently, intact samples of *C. chinensis* were specifically chosen for for qPCR analysis. The related descriptions have been added in the Materials and Methods (Line 513, 517, 553, 555, and 613) and Discussion (Line 327-329) of our revised manuscript.

(6) No specific information is provided regarding the procedure followed for the rescue experiments with burs-α and burs-β (How were they done? Which concentrations were applied? What were the effects?). These important details should appear in the Materials and Methods and the Results sections.

Thanks for your excellent suggestion. For the rescue experiments, the dsRNA of *CcBurs-R* and proteins of burs α-α, burs β-β homodimers, or burs α-β heterodimer (200 ng/μL) were fed together. The concentration of heterodimer protein of *CcBurs-α+β* was 200 ng/μL. The heterodimer protein of *CcBurs-α+β* fully rescued the effect of RNAi-mediated knockdown on *CcBurs-R* expression, while α+α or β+β homodimers did not (Figure 3F). Feeding the α+β heterodimer protein fully rescued the defect in the transition percent and morphological phenotype after *CcBurs-R* knockdown (Figure 4G-4H). We have added the detailed methods of rescued experiments and specific concentrations in the Materials and Methods (Line 561-563), and Results (Line 263) of our revised manuscript.

(7) Pigmentation(a) The protocol used to assess pigmentation needs to be validated. In particular, the following details are needed: Were all pigments extracted? Were pigments modified during extraction? Were the values measured consistent with values obtained, for instance, by light microscopy (which should be done)?

Thanks for your excellent comment. Our protocol for pigment extracted as detailed in *Bombyx mori*, the cuticles were pulverized in liquid nitrogen and then dissolved in 30 milliliters of acidified methanol (Futahashi et al., 2012; Osanai-Futahashi et al., 2012). Thus, all cuticle pigments were dissected and treated with acidified methanol. Pigments were not modified during extraction. The details description have been integrated into the Materials and Methods (Line 630-633) of our revised manuscript.

Futahashi, R., Kurita, R., Mano, H., & Fukatsu, T. (2012). Redox alters yellow dragonflies into red. Proceedings of the National Academy of Sciences of the United States of America, 109(31), 12626–12631. https://doi.org/10.1073/pnas.1207114109

Osanai-Futahashi, M., Tatematsu, K. I., Yamamoto, K., Narukawa, J., Uchino, K., Kayukawa, T., Shinoda, T., Banno, Y., Tamura, T., & Sezutsu, H. (2012). Identification of the *Bombyx* red egg gene reveals involvement of a novel transporter family gene in late steps of the insect ommochrome biosynthesis pathway. The Journal of biological chemistry, 287(21), 17706–17714. https://doi.org/10.1074/jbc.M111.321331

(b) In addition, pigmentation occurs post-molting; thus, the results could reflect indirect actions of bursicon signaling on pigmentation. The levels of expression of downstream pigmentation genes (ebony, lactase, etc) should be measured and compared in molting summer vs. winter morphs.

Thanks for your valuable suggestion. Actually, we already studied the function of some downstream pigmentation genes, including *ebony*, *Lactase*, *Tyrosine hydroxylase*, *Dopa decarboxylase*, and *Acetyltransferase*. The variations in the expression patterns of these genes are closely tied to the molting dynamics of nymphs undergoing transitions between summer-form and winter-form. These findings will put in another manuscript currently being prepared for submission, thus detailed outcomes are not suitable for inclusion in the current manuscript.

(8) L236: "while the heterodimer protein of CcBurs α+β could fully rescue the effect of *CcBurs-R* knockdown on the transition percent (Figure 4G 4H)". This result seems contradictory. If *CcBurs-R* is the receptor of bursicon, the heterodimer protein of CcBurs α+β should not be able to rescue the effect of *CcBurs-R* knockdown insects. How can a neuropeptide protein rescue the effect when its receptor is not there! If these results are valid, then the *CcBurs-R* would not be the (sole) receptor for CcBurs α+β heterodimer. This is a critical issue for this manuscript and needs to be addressed (also in L337 in Discussion).

Thanks for your insightful suggestion. Following the administration of dsCcBur-R to *C. chinensis*, the expression of *CcBurs-R* exhibited a reduction of approximately 66-82% as depicted in Figure 4A, rather than complete suppression. Activation of endogenous *CcBurs-R* through feeding of the α+β heterodimer protein results in an increase in *CcBurs-R* expression, with the effectiveness of the rescue effect contingent upon the dosage of the α+β heterodimer protein. Consequently, the capacity of the α+β heterodimer protein to effectively mitigate the impacts of *CcBurs-R* knockdown on the conversion rate is clearly demonstrated. We have added additional discussion in Line 396-403 of our revised manuscript.

(9) Fig. 5D needs improvement (the magnification is poor) and further explanation and discussion. mi6012 and *CcBurs-R* seem to be expressed in complementary tissues--do we see internal tissues also (see problem under point 2)? Again, the magnification is not high enough to understand and appreciate the relationships discussed.

Thanks for your valuable suggestion. In order to enhance the resolution of the magnified images, we conducted FISH co-localization of miR-6012 and *CcBurs-R* in 3rd instar nymphs and obtained detailed zoomed-in images. As shown in the magnified view of Figure 5D, miR-6012 and *CcBurs-R* appear to exhibit complementary expression patterns in tissues. During the FISH assays, epidermis transparency of *C. chinensis* was achieved via decolorization treatment. Noteworthy observations from Figure 3G and Figure 5E reveal an inverse correlation in the expression profiles of *CcBurs-R* and miR-6012. Consequently, the FISH results distinctly highlight a significant disparity in the expression levels of *CcBurs-R* and miR-6012 within the same tissue. We have added related explanation and discussion in Line 291-293 of our revised manuscript.

(10) The schematic in Fig. 7 is a useful summary, but there is a part of the logic that is unsupported by the data, specifically in terms of environmental influence on cuticle formation (i.e., plasticity). What is the evidence that lower temperatures influence expression of miR-6012? The study measures its expression over life stages, whether with an agonist or not, over a single temperature. Measuring levels of expression under summer form-inducing temperature is necessary to test the dependence of miR-6012 expression on temperature. Otherwise, this result cannot be interpreted as polyphenism control, but rather the control of a specific trait.

Thanks for your great suggestion. We actually conducted the assessment of miR-6012 expression at specific time intervals (3, 6, 9, 12, 15 days) under different temperatures of 10°C and 25°C. As depicted in Figure 5E, the expression levels of miR-6012 were notably reduced at 10°C compared to 25°C. Additionally, the evaluation of agomir-6012 expression level of *C. chinensis* under 25°C conditions at various time points (3, 6, 9, 12, 15 days) revealed no significant changes. Hence, we suggest that the impact of miR-6012 on the seasonal morphological transition is influenced upon temperature.

**Recommendations for the authors:**
The authors report a novel role of Bursicon and its receptor in regulating the seasonal polyphenism of *Cacopsylla chinensis*. They found that low temperature treatment (10°C) activated the Bursicon signaling pathway during the transition from summer-form to winter-form, which influences cuticle pigment content, cuticle chitin content, and cuticle thickness. Moreover, the authors identified miR-6012 and show that it targets *CcBurs-R*, thereby modulating the function of Bursicon signaling pathway in the seasonal polyphenism of *C. chinensis*. This discovery expands our knowledge of multiple roles of neuropeptide bursicon action in arthropod biology. However, the manuscript does have several major weaknesses, described under "Public review", which the authors need to address.Major issues:(1) L152-154 Fig S2E and S2F: Bursicon has been shown to be expressed in the CNS in a specific set of neurons. For example, In the larval CNS of *Manduca sexta*, bursicon expression is restricted to the subesophageal ganglion (SG), thoracic ganglia, and first abdominal ganglion. Pharate pupae and pharate adults show expression of this heterodimer in all ganglia. In *Drosophila* larvae, expression of a bursicon heterodimer is confined to abdominal ganglia. The additional neurons in the ventral nerve cord express only burs. In pharate adults, bursicon is produced by neurons in the SG and abdominal ganglia. I am wondering where bursicon subunits are expressed in the *C. chinensis* CNS? Since the authors have the antibodies, it would be useful to include immunocytochemical staining of bursicon alpha and beta in the CNS. The qPCR results from head or other tissues (Fig S2E and S2F) is not the most informative way to document localization of gene expression. Regarding the qPCR results, they show that the cuticle and the fat body express *CcBurs-α* and *CcBurs-β*. Can the authors confirm this unexpected results independently?

Thanks for your insightful comment. In this study, we did not directly used antibodies targeting bursicon subunits, instead, the bursicon subunits along with a histidine tag were integrated into the expression vector pcDNA3.1 using homologous recombination. The experimental procedures were executed as follows: initially, the histidine tag was fused to the pcDNA3.1-mCherry vector through homologous recombination to generate the recombinant plasmid pcDNA3.1-his-mCherry. Subsequently, the amino acid sequences of the two bursicon subunits were introduced into the pcDNA3.1-his-mCherry vector via homologous recombination to produce the recombinant plasmids pcDNA3.1-CcBurs-α-his-mCherry and pcDNA3.1-CcBurs-β-his-mCherry. Finally, the P2A sequence was incorporated into the vector using reverse PCR to yield the recombinant plasmids pcDNA3.1-CcBurs-α-his-P2A-mCherry and pcDNA3.1-CcBurs-β-his-P2A-mCherry. Consequently, the bursicon subunits, along with the histidine tag, were capable of generating fusion proteins with the histidine tag. Western blot analysis was conducted using antibodies targeting the histidine tag, enabling the detection of histidine expression, which corresponds to the expression of the bursicon subunits. However, they are not suitable to conduct the in vivo immunocytochemical staining of bursicon alpha and beta in the CNS.

Due to the diminutive size of the *C. chinensis* nymphs, dissection of the central nervous system (CNS) was unfeasible, precluding specific assessment of bursicon expression in the CNS. Prior literature has documented the expression of bursicon subunits in the epidermis and fat body of *C. chinensis*. Studies suggest that bursicon subunits not only play a role in the melanization and sclerotization processes of insect epidermis but also have significant roles in insect immunity (An et al., 2012). The presence of bursicon subunits in the epidermis, gut, and fat body of *C. chinensis* may indicate their crucial roles in the immune functions of these tissues. Further investigation is required to elucidate the specific immune functions they perform, hinting at the potential expression of these bursicon subunits in these two tissues.

An, S., Dong, S., Wang, Q., Li, S., Gilbert, L. I., Stanley, D., & Song, Q. (2012). Insect neuropeptide bursicon homodimers induce innate immune and stress genes during molting by activating the NF-κB transcription factor Relish. PloS one, 7(3), e34510. https://doi.org/10.1371/journal.pone.0034510

(2) L222: "*CcBurs-R* is the Bursicon receptor of *C. chinensis*". Is this statement supported by affinity binding assay results?

Thanks for your excellent suggestion. We employed a fluorescence-based assay to quantify calcium ion concentrations and investigate the binding affinities of bursicon heterodimers and homodimers to the bursicon receptor across varying concentrations. Our findings suggest that activation of the receptor by the burs α-β heterodimer leads to significant alterations in intracellular calcium ion levels, whereas stimulation with burs α-α and burs β-β homodimers, in conjunction with Adipokinetic hormone (AKH), maintains consistent intracellular calcium ion levels. Consequently, this research definitively identifies *CcBurs-R* as the bursicon receptor. For further details, please refer to the Materials and Methods (Lines 493-504), Results (Lines 231-239), and Discussion (Lines 377-384) of our revised manuscript.

(3) L245 Figure 4I-4J: Since knockdown of bursicon and its receptor cause a decrease pigment accumulation in the cuticle, it would be useful to examine 1-2 rate limiting enzyme-encoding genes in the bursicon regulated cuticle darkening process if possible (as was done for genes involved in cuticle thickening).

Thanks for your excellent comment. Following the further study, a thorough analysis was conducted to evaluate the impact of bursicon and its receptor on the expression levels of *Lactase*, *Tyrosine hydroxylase*, *Dopa decarboxylase*, *Acetyltransferase*, and the effects of RNA interference targeting these genes on the seasonal morphological transition. The findings underscored their role in the bursicon-mediated cuticle darkening process. However, as this section is slated for inclusion in an upcoming manuscript intended for submission, it is deemed unsuitable for incorporation into the current manuscript.

Minor issues:(1) L75 "stronger resistance (Ge et al., 2019; Tougeron et al., 2021)". Stronger resistance to what? Stronger resistance to environmental stress or weather condition? Please clarify.

Thanks for your excellent suggestion. We have changed the statement to “stronger resistance to weather condition” in Line 75 of our revised manuscript.

(2) L132 Figure 1A and 1B: Bursicon sequence was first identified and functionally characterized in *Drosophila melanogaster*: is there any reason why *Drosophila* bursicon sequences were not included in the comparison?

Thanks for your excellent comment. We have added the sequence of *Burs-α* and *Burs-β* of *D. melanogaster* in the sequence alignment results of Figure 1A and 1B of our revised manuscript.

(3) Although the authors clearly identify and validate the function for the bursicon genes and its receptor's, there is no mention of whether duplicates of this gene are also present in the pear psyllid. This has been known to happen in otherwise conserved hormone pathways (e.g., insulin receptor in some insects), so a formal check of this should be done.

Thanks for your excellent comment. As shown in Figure S2A-S2B and 3B, there are two bursicon subunit genes and only one bursicon receptor gene in our selected insect species, for examples *Drosophila melanogaster*, *Diaphorina citri*, *Bemisia tabaci*, *Nilaparvata lugens*, and *Sogatella furcifera*. In our transcriptome database of *C. chinensis*, we also only identified two bursicon subunit genes and only one bursicon receptor gene.

(4) Line 41: Here, as in the title, "fascinating" is a subjective judgement that does not improve a study's presentation.

Thanks for your great comment. We have changed "fascinating" to "transformation" in Line 41 and also revised the title of our revised manuscript.

(5) Line 44: What makes some fields "cutting-edge" and others not?

Thanks for your excellent suggestion. The expression of "in cutting-edge fields" has been deleted in Line 44 of our revised manuscript.

(6) Line 97: This is a peculiar choice of reference for the concept of slower development in cold temperatures. The concept of degree-days and growth rates is old and widespread in entomology.

Thanks for your insightful comment. The reference of Nyamaukondiwa et al., 2011 in Line 95 has been deleted in our revised manuscript.

(7) Lines 149-150: What justifies the assumption that higher levels of expression mean a more important role? This gene might be just as necessary for development of the summer form, even if expressed at lower levels.

Thanks for your excellent suggestion. This sentence has been revised to “Increased gene expression levels may potentially contribute to the transition from summer-form to winter-form in *C. chinensis.*” in Line 168-169 of our revised manuscript.

(8) The blue arrow in Fig. 7 is confusing.

Thanks for your excellent suggestion. In Figure 7, the blue arrow represents the down-regulated expression of miR-6012. We have added a description about the blue arrow in Figure 7 of our revised manuscript.